

# Characterisation of ozone deposition to a mixed oak-hornbeam forest. Flux measurements at 5 levels above and inside the canopy and their interactions with nitric oxide

Angelo Finco[1,] Mhairi Coyle[2], Eiko Nemitz[2], Riccardo Marzuoli[1], Maria Chiesa[1], Benjamin Loubet[3], Silvano Fares[4], Eugenio Diaz-Pines[5], Rainer Gasche[6] and Giacomo Gerosa[1,*]

[1] Dipartimento di Matematica e Fisica, Università Cattolica del S. C., Brescia, Italy

[2] Centre for Ecology & Hydrology, Bush Estate, Penicuik, United Kingdom

[3] Institut National de la Recherche Agronomique, Thiverval-Grignon, France

[4] Council for Agricultural Research and Economics, Research Centre for Forestry and Wood, Arezzo, Italy

[5] Institute of Soil Research, University of Natural Resources and Life Sciences (BOKU); Vienna, Austria.

[6] Institute of Meteorology and Climate Research Atmospheric Environmental Research (IMK-IFU), Garmisch-Partenkirchen, Germany

*Correspondence to*: Giacomo Gerosa (giacomo.gerosa@unicatt.it)

**Abstract.** In the framework of the European FP7 project ECLAIRE a joint field campaign was run in Marmirolo, in the northern part of the Italy, one of the most polluted areas in Europe due to intense industrial activities and peculiar climate
conditions promoting high ozone formation by photochemical reactions involving nitrogen oxides ($NO_x$) and volatile organic compounds. The studied ecosystem is a mixed oak-hornbeam forest and the aim of this field campaign was to investigate the processes regulating the gas exchange between the forest and the atmosphere with a focus on ozone flux measurements and the interaction with reactive gases. Measurements were run on a 40 m tower equipped with sonic anemometers and fast ozone analyzers for eddy covariance at five different heights: one at canopy height, two above canopy and two below
canopy. NO fluxes were measured above canopy and $NO_x$ fluxes were measured with a dynamic chamber system at the soil level. Ozone fluxes measured at different levels above the canopy showed a good agreement between each other, while fluxes at 24 m were surprisingly higher than the above ones. In this paper we discuss the possible reasons for this discrepancies shedding light on the role of $NO_x$ and of the coupling between forest and atmosphere will be explained. A partition of the ozone fluxes will be shown too to identify the most relevant sinks in the soil-plant continuum.



## 1 Introduction

Ozone (O₃) had been widely documented as the most dangerous pollutant for plants (Ashmore, 2005; Matyssek and Innes, 1999; Matyssek et al., 2012), and damages to vegetation had been observed from the cellular up to ecosystem level (e.g., Dizengremel et al., 2013, Marzuoli et al. 2008). Prompted by its phytotoxicity, the deposition of O₃ to forest ecosystems had

been widely studied over the last 20 years (e.g. Padro, 1996; Cieslik, 1998; Lamaud et al., 2002; Mikkelsen et al. 2004, Gerosa et al. 2005, Gerosa et al. 2009, Launianen et al., 2013), thanks also to the development of fast ozone analysers, suitable for eddy covariance measurements.

Measurements carried out in the 1990´s were usually short-term field campaigns, while more recently campaigns had extended their observation periods and therefore led to a better understanding of the processes controlling ozone deposition

(e.g. Mikkelsen et al. 2004; Gerosa et al. 2008; Fowler et al., 2009, Rannik et al., 2012; Zona et al., 2014; Fares et al. 2014; Clifton et al., 2016).

The fate of the ozone deposition below canopy is yet to be completely understood. Very few studies (e.g. Launianen et al., 2013; Fares et al. 2014, Dorsey et al. 2004) have measured ozone fluxes below the canopy of a forest. While the first two papers were more focused on the validation of deposition models for all the possible pathways of ozone deposition, the latter

was more focused on the role of soil NO emission in the ozone flux dynamics.

A field campaign was conducted in 2012 in the context of the European FP7 project ECLAIRE ("Effects of Climate Change on Air Pollution Impacts and Response Strategies for European Ecosystems") in Italy's Po Valley, one of the most polluted areas in Europe. The aim was to contribute to the understanding of the dynamics of ozone fluxes and concentrations above and within forest canopies, as well as its interaction with NOₓ exchange. A partition of the ozone fluxes into the different

sinks (upper and lower crown layer, chemical sinks, understory and soil) of the ecosystem was performed to support our study on possible drivers of the ozone fluxes along with their interactions with meteo-climatic controls.

Thanks to an extremely detailed dataset obtained from five eddy covariance systems for ozone along a vertical profile on a tower and concurrent measurement of NOₓ fluxes above the canopy and at the forest floor, another aim of this work is to test the capacity of existing deposition models to predict intra-canopy dynamics involving ozone reactions with NOₓ and VOC.

For this sake, data from a joint field campaign, conducted in the context of the European FP7 project ECLAIRE ("Effects of Climate Change on Air Pollution Impacts and Response Strategies for European Ecosystems"), are analysed. This field campaign took place in the summer of 2012 in a forest in the Po valley, in northern Italy, where five eddy covariance systems for ozone were deployed along a vertical profile on a tower: two outside the canopy, one at top canopy level and two below canopy. At the same time, NOₓ fluxes were measured above the canopy and at the forest floor.

Simultaneous flux measurements of volatile organic compounds, particles and inorganic gases containing nitrogen and sulphur are reported elsewhere (Acton et al., 2016; Schallhart et al., 2016).



## 2 Material and methods

### 2.1 Site characteristics

Measurements were performed at the Bosco Fontana reserve (45°12'02"N, 10°44'44"E; elevation 25 m a.sl.) located in Marmirolo near Mantua, Italy. The measuring site is a mixed oak-hornbeam forest, a typical climax ecosystem of the area

and it is located just in the middle of the Po Valley, one of the most polluted areas of Europe. The forest forms part of a 233 ha nature reserve classified as a Site of Communitarian Importance and Special Protection Zone (IT20B0011) and it is part of the Long-Term Environmental Research (LTER) network.

The dominant tree layer is composed of hornbeam (*Carpinus betulus*, 40.45 % of the total surface of the reserve), oak (*Quercus robur*, 17.09 %), red oak (*Quercus rubra*, 9.65 %) and Turkey oak (*Quercus cerris*, 7.06 %) (Dalponte et al.,

2007). Some species (*Acer campestre*, *Prunus avium*, *Fraxinus ornus and oxycarpa*, *Ulmus minor, and Alnus glutinosa* along the small streams) are present but they account for no more than 3% of the total surface.

The dominated tree layer is made up of *Corylus avellana*, *Sambuscus* spp, *Cornus mas*, *Crataegus oxyacantha and monogyna and Sorbus torminalis* . A thick nemoral layer of butcher's broom (*Ruscus aculeatus*, L) is also present.

The average height of the canopy is 26 m and the average single-sided leaf area index (LAI), measured by a canopy structure

meter (LAI2000), averaged 2.28 $m^2\ m^{-2}$, with a maximum of 4.22 $m^2\ m^{-2}$.

The soil is a Petrocalcic Palexeralf, loamy skeletal, mixed, mesic (Campanaro et al. 2007) according to the USDA classification. The soil depth is 1.5 m with petrocalcic hardened layer between 0.80 and 1 m below the ground; this layer was formed after the gradual deepening of the water table.

The climatic characteristics are typical of the Po Valley, with humid and hot summers (Longo, 2004). The mean annual

temperature is 13.2°C (period 1840-1997, Bellumé et al., 1998) and July is the hottest month (24.6°C).

The most frequent wind directions are generally from E and NE, in particular in spring and summer.

### 2.2 Measuring infrastructure

A 40 m tall scaffold walk-up tower was constructed inside the forest (45°11'52.27"N, 10°44'32.27"E), with a measuring fetch ranging between a minimum of 390 m in the S direction and a maximum of 1440 m in the NE direction.

The infrastructure was equipped with instrumentation for four different kinds of measurements: fluxes of energy and matter ($O_3$, $NO_x$, $CO_2$, $H_2O$) with the eddy covariance technique, soil flux of $O_3$ and $NO_x$ with dynamic chambers, vertical profiles of gas concentrations ($O_3$, $NO_x$) and air temperature and humidity, and additional meteorological and agrometeorological measurements (solar radiation, rainfalls, soil temperature, soil heat fluxes and soil water content).

### 2.3 Eddy covariance measurements of matter and energy fluxes

Four sonic anemometers (see **Table 1** for models) were placed on the tower at four different heights: 16 m, 24 m, 32 m and 41 m. A fifth one was installed at 5 m a.g.l. on a pole, 10 m away from the tower. At the top tower level an open path



infrared gas analyser (Model 7500, LI-Cor, USA) was also installed to measure the concentrations of water vapour and carbon dioxide, and at each of the five sampling heights a fast ozone instrument was installed to measure ozone vertical fluxes.

All the fast ozone instruments (see **Table 1** for models) were based on the reaction between ozone and a cumarine-47 target

which has to be changed after some days because its sensitivity declines exponentially with time (Ermel et al., 2013). Three fast ozone instruments (two COFA and the ROFI) broadly followed the design of the GFAS instrument (Güsten and Heinrich, 1996) equipped with a relatively big fan (about 100 L min$^{-1}$) which resulted in a faster consumption of the cumarine target than the other two instruments: a prototype developed by the National Oceanic and Atmospheric Administration (NOAA, FROM, Bauer et al., 2000) and the commercial Fast Ozone Sensor (FOS, Sextant, NZ), both of

which utilise a small membrane pump (2.5 L min$^{-1}$). For this reason the ozone targets were changed every 5 days for COFA and ROFI and every 10 days for the FROM and the FOS. In both cases the ozone targets were pre-conditioned just before use by exposing them to a concentration of 100 ppb of ozone for two hours.

Above-canopy fluxes of nitric oxide (NO) were measured at 32 m by means of a CLD780TR fast analyzer (Ecophysics, CH) based on the chemiluminescence reaction between $O_3$ and NO. The air to be analyzed was drawn from 32 m through a 3/8 ID

Teflon tube main line at 60 L min$^{-1}$ to the analyzer placed at the bottom of the tower. The analyzer was sub-sampling at 3 L min$^{-1}$ from the main sampling line. The CLD780TR was calibrated with an 80 ppb standard produced using a dilution system (LNI 6000x, S) and a standard NO cylinder (18 ppm), at the beginning of the experiment and then weekly.

All the fast instruments and the sonic anemometers were sampled at 20 Hz through a customized LabVIEW (National Instruments, IRL) program and data were collected and stored in hourly files.

**2.4 Soil NO, NO$_2$ and O$_3$ flux measurements**

Fluxes and concentrations of NO, NO$_2$ and O$_3$ at the soil-atmosphere interface were determined by use of a fully automated measuring system as described in detail elsewhere (Butterbach-Bahl et al., 1997; Gasche and Papen, 1999; Rosenkranz et al., 2006; Wu et al., 2010). Briefly: five dynamic measurement chambers and one dynamic reference chamber were installed at the site. Dimensions of the chambers were: 0.5 m x 0.5 m x 0.15 m (length x width x height). In contrast to the measuring

chambers, the reference chamber was sealed gastight against the soil surface using a plate made of perspex. Time resolution for flux measurements was 1 hour. Every chamber was closed and measured for 6 minutes, and before every sampling of a measuring chamber the reference chamber was sampled, resulting in a measuring cycle of 60 minutes. During sampling, the air from the chambers was sucked at a constant rate of 50 L min$^{-1}$ and transported via PTFE tubing (inner diameter: 10 mm, length 20 m) to the analyzers. NO and NO$_2$ concentrations were determined using a chemiluminescence detector CLD 770

AL equipped with a photolytic converter (Models CLD 770AL and PLC 760, Ecophysics, CH), and O$_3$ concentrations were determined using an UV ozone analyzer (mod. TE49C, Thermo Environmental Instruments, USA). Corrections for initial concentrations of NO, NO$_2$ and O$_3$ at the outlet of each chamber and calculation of fluxes of NO and NO$_2$ was performed according to Butterbach-Bahl et al. (1997). Calibration of the chemiluminescence detector was performed weekly using 40




ppb NO in synthetic air produced by dilution of standard gas (4 ppm NO in $N_2$) with synthetic air (80% $N_2$, 20% $O_2$) using a multi gas calibrator (mod. 6100, Environics, USA). Efficiency of photolytic conversion of $NO_2$ to NO was determined at least weekly as described in detail by Butterbach-Bahl et al. (1997).

### 2.5 Vertical profile of $O_3$ and $NO_x$ concentrations and air temperature and humidity

A computer driven system of Teflon tubes and solenoidal Teflon valves was used to characterize the vertical concentration profile of $O_3$ and $NO_x$ above and within the canopy at 6 heights:  5 m, 8 m, 16 m, 24 m, 32 m and 41 m. The air samples drawn through 3/8 ID Teflon tubes (all of them 50 m long) from each level by a 30 L $min^{-1}$ pump were sequentially sent to an UV ozone photometer (mod. 49C, Thermo Scientific, USA) and to a $NO_x$ chemiluminescence analyser (mod. 42C, Thermo Environmental Instruments, USA) lodged in an air-conditioned container at the bottom of the tower. Both analyzers

were sub-sampling at 2 L $min^{-1}$ out of the 3/8 ID sampling lines.

All the tubes were insulated and continuously purged. Each level was sampled for 4 minutes after 1 minute wait to let the analyzers stabilize and then concentration data were recorded each half an hour with a customized LabVIEW program (National Instruments, USA).

The $O_3$ gradient analyzer was calibrated against a reference photometer before and after the field campaign and no

significant deviation from the first calibration was observed. The $NO_x$ analyzer was calibrated with the same procedure described above for the Ecophysics CLD780TR at the beginning of the experiment and then weekly.

Additional $O_3$ and $NO_x$ concentrations at 0.15 m were also available from the soil chambers system previously described.

The ozone concentrations at 5 m, 16 m, 24 m, 32 m and 41 m were also used as absolute ozone reference for the fast ozone instruments all of which change sensitivity sufficiently fast to require constant calibration against a slow response absolute

instrument.

Five temperature and relative humidity probes (mod. HMP45,Vaisala, Finland) were installed, at the higher tower levels (16 m, 24 m, 32 m and 41 m) and, additionally, at 11 m. All these probes were connected to a data logger (CR23x, Campbell Sci., USA), sampled once per minute and stored as half an hour averages.  Two additional temperature measuring points (PT100, Campbell Scientific, USA), were available at 1.5 m and 0.15 m a.g.l. and data were collected with the same personal

computer used for the control of the dynamic chamber system.

### 2.6 Additional meteorological and agrometeorological measurements

On the top of the tower a net radiometer NR-lite (Kipp & Zonen, NL), a BF5 sunshine sensor for total and diffuse PAR (Delta-T Devices, UK), a PTB101B barometer (Vaisala, Finland) and a rain gauge (mod. 52202, Campbell Scientific, USA) were mounted.

Several soil probes were deployed in the soil 20 m from the bottom of the tower: four reflectometers for soil water content (TDR mod 616, Campbell Scientific, USA), four soil heat flux plates (mod. HFP01SC, Hukseflux, NL) and four soil





temperature probes (PT100, GMRstrumenti, I). All these sensors were connected to a data logger (CR13x, Campbell Scientific, USA), sampled once per minute and stored as half an hour averages.

### 2.7 Measuring period

The measuring campaign began on 12[th] June and ended one month later, on 11[th] July 2012. From the 12[th] June to the 23[rd]
June three fast ozone instruments (ROFI, FROM and one of the two COFA samplers) were all placed above the canopy at a height of 32 m in order to compare them and to characterize their performances ("Intercomparison period"). The COFA installed at the top of the tower started its measurements on 12[th] June and was not moved to level 32 m for the intercomparison because it was already compared with the second COFA before the campaign.

The intercomparison exercise allowed the agreement between the three instruments to be verified, and the average relative
standard deviation was below 10%. Considering the intrinsic variation due to the different behaviour of individual cumarine targets no (systematic) correction was applied. The sextant analyzer, the one employed at 5 m, was calibrated after the field campaign against one of the two COFA. Also in this case, no significant deviation was observed and no corrections were applied.

On 24[th] June each fast ozone instrument was moved to a different level (**Table 1**) to begin the flux profile measurements
which ended the 11[th] July ("Flux Profile period").

The FOS installed at 5 m was checked after the field campaign by running it in parallel with the COFA previously used at 32 m in the intercomparison period.

### 2.8 Data processing

The flux measurement technique adopted here was the eddy covariance (e.g. Foken, 2008), which states that fluxes are equal
to the covariance between the vertical wind component and the scalar of interest (Arya, 2001). An averaging period of 30 minutes was chosen for the calculation of the covariances.

The following procedures were applied for the flux calculation using home-made software written in Borland Delphi.

*Despiking*. The data series were divided into 2 minutes sub-series and for each of them block average and standard deviation
were calculated. Spikes were identified as the instantaneous data that exceeded the average of each sub-series for more than 3.5 times the standard deviation, as proposed by Vickers and Mahrt (1997). Spikes were removed from the series and the data were then gap-filled by a linear interpolation.

*Rotations*. Two axis rotations were applied to the instantaneous wind components to align $u$ with the mean flow over the averaging period: the first rotation aligned the horizontal wind to the 30 minutes average $u$ component (this rotation forces
$\bar{v} = 0$), and the second one to rotate the $xy$ plane in order to zero the 30 minutes average vertical component of the wind ($\bar{w} = 0$) (McMillen, 1988; Wilczak et al., 2001). These rotations corrected the little imperfections in the vertical alignment





of the sonic anemometers and prepare the data for flux calculations. Samples with a second rotation (vertical tilt) angle greater than 15° were discarded.

*Linear detrending.* The fluctuations of each parameter (e.g. w', T', O₃') were calculated as the differences of each instantaneous value from the best linear fit (minimum square) of the considered time series in each half an hour data (Lee at al., 2004).

*Time-lag determination.* Ozone fluxes were calculated using a fixed time-lag between the vertical wind time series and the ozone concentration one. For each fast instrument the time lag which maximized the cross-covariance function between the vertical component of the wind and the ozone concentrations were identified and the more frequent lag was used in the calculations for every half an hour.

*Elimination of the kinematic fluxes below the error threshold.* The error threshold was quantified for each semi-hourly data series by following the methodology proposed by Langford et al. (2015). The standard deviation of the auto-correlation function was calculated for each half hour data chunk, with lags ranging between 30 and 60 seconds from the characteristic time lag of each instrument. kinematic fluxes lower than three times the standard deviation (relating to the 95[th] percentile) of were discarded.

*Frequency loss correction.* The frequency loss correction factors for the different fast ozone instruments were identified by calculating the cospectra between w and O₃ and by performing an ogive analysis according to Rummel et al. (2007). The ogives are the cumulative co-spectra and show how much of the flux is carried by frequencies up to a certain value. Ogives were calculated and normalized to unity both for ozone and sensible heat fluxes. The correction factor to be applied to each fast ozone analyzer was obtained by scaling the ozone flux ogive to match the sensible heat ogive at low frequencies (up to 0.0065 Hz), and then calculating the ratio between the sensible heat and the ozone ogives at high frequencies.

*Schotanus and WPL corrections.* Fluxes of sensible heat (H), latent heat (LE) and trace gases were corrected for air density fluctuations. The formulation adopted for the correction of H was the one proposed by Schotanus et al. (1983) while the formulation used for LE and trace gases was the one proposed by Webb et al. (1980).

*Calculation of fluxes in physical units.* Fast ozone concentration data –acquired as voltages- and fast NO concentration data –acquired as counts per seconds- required additional processing to calculate quantitative fluxes in physical units. First, for all the fast ozone instruments the target zero $V_0$ (Muller et al., 2010) – i.e. the output voltage at null ozone concentration– was identified for each cumarine target employed. Then the ozone fluxes were calculated by the following equation (Muller et al., 2010):

$$F_{O3} = \frac{\overline{w'V'}}{\overline{V} - V_0} \, C_{O3} \qquad\qquad (1)$$

where $\overline{w'V'}$ is the covariance between the vertical wind component and the raw output voltage of the fast ozone instrument, $\overline{V}$ is the average output voltage of the instrument in each half an hour, $V_0$ is the zero target identified for the considered half an hour, and O₃ is the ozone concentration measured by the reference ozone analyzer averaged over the same period. The data of the two hours following each target change were excluded in order to let the target sensitivity to stabilize after the





target installation.

Similarly, NO fluxes were calculated using the following equation:

$$F_{NO} = \frac{\overline{w'cps'}}{\overline{cps} - cps_0} S_{NO} \tag{2}$$

where $\overline{cps}$ (counts per seconds) is the NO raw data measurement averaged in each half an hour, and $cps_0$ and $S_{NO}$ are the offset and the sensitivity of the analyser determined by calibration. $Cps_0$ ranged from 1000 to 2500 cps while $S_{NO}$ ranged from 10000 to 12000 cps / (µmol m$^{-3}$).

*Ozone storage*. Ozone fluxes measured by eddy covariance were corrected for the ozone storage. The ozone storage is the temporal variation of the vertical ozone profile below the measuring point located at the height $z_m$. It does not represent a true ozone removal or production process, but only a temporary accumulation of ozone in the air column below the measuring point or a temporary ozone release out of the same air column. For a non-reactive tracer the proof of it is that the storage integrated over a whole day is null. The calculation of the ozone storage is necessary for a proper determination of the ozone deposition processes.

The correction of the ozone fluxes for the storage was made by means of the following equation (Rummel et al., 2007)

$$F_{StorO_3} = F_{O_3} + \frac{\partial}{\partial t} \int_0^{z_m} O_3(z) dz \tag{3}$$

where $F_{StorO_3}$ are the ozone fluxes corrected by storage, $F_{O_3}$ are the measured ozone fluxes obtained with the **Eq. 1**, and the second term on the right represents the ozone storage term. For a reactive tracer like ozone some of the stored gas may be destroyed by reaction with NO and potentially VOCs before it can be re-released to the air space above and Eq. (3) must be considered an approximation. A fully resolving 1D chemistry and exchange model would be required to quantify the effect of chemistry on the storage term more fully.

*Stationarity check*. Finally the stationarity of each 30 minutes sample was verified following the methodology by Foken and Wichura (1996) and the non-stationary data were discarded.

## 3 Results

### 3.1 Microclimate

Significant rainfalls had cooled air before the beginning of the field campaign so that, air temperature increased significantly in the first days and after that remained stable (Figure 1a). The average temperature at the top of the tower was 25.9 °C, while the lowest average temperature (23.1 °C) was recorded at 0.15 m. The maximum temperature during the whole period was 36.2 °C which was observed at the top of the canopy (24 m).

On average the temperature minimum was observed during night around 3:00 (always CET hereafter) (Figure 1b) for the levels from 11 m to 24 m and one hour later for the other ones, with values ranging from 19 °C to 21 °C. Only two significant rainfall events occurred in the final part of the campaign accounting for a total of 108 mm of rain, but these did





not affect significantly the air temperature (Figure 1a). In general most of the days were sunny (only three days were partially cloudy) and humid, with nighttime peaks of relative humidity up to 80% and diurnal minima around 40%. Specific humidity $q$ ranged, on average, between 10 and 13 $g_{H_2O}/kg_{air}$ (Figure 1c). Below canopy levels ($\leq$ 16 m) showed higher $q$ than the above canopy levels early in the morning, around 6:00, and from 13:00 to 21:00, while the top-crown level (24 m)

showed the lowest $q$ values from 4:00 to 16:00. Specific humidity showed close agreement amongst the above canopy levels, with slightly higher values at 41 m.

The wind blew mostly from the East or West, with about 50% of the data in these directions (Figure 1d), whilst the N and S directions accounted for 12% of the data and the intermediate directions accounted for less than 20% of the data. The diurnal wind intensity at 41 and 32 m was on average around 2 and 1.5 m s$^{-1}$ respectively (Figure 1e), and the wind intensity was

slightly greater during night than during daytime, with nearly 1 m s$^{-1}$ more at 41 m and 0.5 m s$^{-1}$ at 32 m. The three lower levels showed very low intensity, below 0.5 m s$^{-1}$, with only a minor increase during the day.

The friction velocity ($u^{*}$) at the two upper levels above the canopy showed a very similar behaviour (Figure 1f) but $u_*$ was slightly higher at 32 m (+6%). Diel maxima of $u_*$ were about 0.5 m s$^{-1}$ occurring between 9:00 and 13:00, after which they suddenly decrease of nearly 20% and then gradually decreased. The minimum (0.13 m s$^{-1}$) was observed around 20:00, after

that a quite irregular behaviour during the night was observed with values between 0.2 and 0.3 m s$^{-1}$. The in-canopy measurements of friction velocity at the lowest three levels were significantly lower than the above canopy ones: 24 m and 5 m measurements were less than 50% of the two upper levels and 16 m measurements were around 70% less than above canopy levels. The diurnal maxima at noon were 0.25 m s$^{-1}$ at 24 m, 0.18 m s$^{-1}$ at 16 m and 0.26 m s$^{-1}$ at 5 m and during the night the friction velocity showed a flatter trend with values below around 0.1 m s$^{-1}$ at 5 m and around 0.05 m s$^{-1}$ for the two

others levels.

### 3.2 Profiles of temperature, heat fluxes and atmospheric stability

Following sunrise, early in the morning, the heating of the top part of the canopy developed a thermal inversion in the forest with the ceiling at the top of the canopy (level 24 m) and the base at ground level (Figure 2a).

Above the canopy temperature gradients were strongly super-adiabatic from 4:00 to 17:00, however it should be noted that

heat transfer increased significantly only when friction velocity increased.

During the morning, the gradual heating of the canopy extended down to the whole crown of the dominant vegetation layer, reaching its maximum value at noon. As a consequence the inversion ceiling was lowered to the bottom part of the tree crowns (16 m). But already from 2 pm the air layers in the middle of the trunk space started to cool and the inversion ceiling gradually reached the top of the canopy. By 6 pm the top of the canopy had cooled sufficiently for and the above-canopy

atmosphere to become stable, which then attenuated overnight, without ever disappearing (Figure 2b).



The presence of an inside canopy thermal inversion is confirmed also by the measured sensible heat fluxes (Figure 3). Above the canopy the heat fluxes were strongly upward during the day. However, the sensible heat fluxes at 32 m were about 20% larger than those at 41 m.

In the upper part of the crown (24 m), sensible heat fluxes were less than half the above canopy ones in the central part of the
day. On the contrary, the heat fluxes at 16 m and 5 m were almost always zero or negative (directed downwards). In relation to the strengthening of the thermal inversion in the afternoon, it is worth noting that the downward heat fluxes peaked at 2 PM at 5 m, two hours later at 16 m and four hours later at 24 m.

However the forest released most of the energy as latent heat with a peak around 300 W m$^{-2}$ at midday and with very small nighttime values.

Above the canopy the atmosphere was nearly always unstable during the day, while below canopy it was mostly stable, as shown in Figure 4. At the top canopy level (24 m) the most frequent condition in the central hours of the day was strong instability because of the canopy heating due to the radiation. Remarkably, stable conditions at this level strengthened from 3 pm to 7 pm just during the inversion.

Inside the canopy (16 m) the atmosphere was mainly stable or very stable. In particular from 14:00 to 19:00 the inside
canopy air was almost always very stable, as it happened for a shorter period in the morning from 6:00 to 8:00. During the night the atmosphere was mainly stable or very stable above canopy, while at 24 m and 16 m some nocturnal instability were observed, this latter might be due to numerical artifacts because the sensible heat fluxes were close to zero. A similar explanation can be used also for the stability class distribution at 5 m, for which some instability was observed. In any case, stable condition was the most frequent situation observed at that level.

**3.3 Ozone concentrations profiles**

Ozone concentrations above the canopy (41 m and 32 m) showed the typical bell-shaped diurnal pattern, with a maximum around 80 ppb at 14:00 and minimum around 25 ppb at 4:00 (**Figure 5**). The concentrations decreased slightly throughout the canopy (-9% between 32 m and 5 m), while there was a remarkable reduction near the ground (-72% between 32 m and 0.15 m). At ground level, average ozone concentrations never exceeded 27 ppb. It is worth noticing the second (relative)
minimum observed at 16:00 at the lowest level; this minimum is in agreement with a slight reduction in the ozone concentrations observed in the upper levels inside the canopy (from 5 m to 24 m). These features can be better observed considering the vertical variations in Figure 6a and Figure 6b. During the night the in-canopy gradient of ozone was negligible, but from early morning a negative gradient rapidly developed and remained almost constant (around 0.2 ppb m$^{-1}$) during the daylight hours, except in the afternoon. The slope of this gradient increased in the afternoon: at 16:00 from 8 m to
32 m (around 0.5 ppb m$^{-1}$) and at 18:00 but only from 24 m to 32 m (around 0.8 ppb m$^{-1}$). Another peculiarity emerged from 13:00 to 15:30, when the ozone concentration just above the canopy (32 m) was on average higher (by 2.0 to 3.8 ppb) than above (41 m); moreover, in the same period, also the 24 m ozone concentration was higher than the one measured at 41 m (from 1.2 to 2.5 ppb).





### 3.4 Ozone fluxes profile

Ozone fluxes were corrected for the storage in the air layers below each measuring point. The magnitude of these corrections was not negligible and they were higher in the morning and in the evening (Figure 7) when the air layers in the trunk space are respectively refilling and emptying of ozone. Considering the whole 41m height air column, the greatest storage

correction was nearly +5 nmol m$^{-2}$ s$^{-1}$ in the morning, while in the evening it was about -4 nmol m$^{-2}$ s$^{-1}$, the integrated value over the day was null.

Ozone fluxes showed a regular behaviour with almost always negative values except for some positive peaks during night or during the transition between night and day, in particular in the lowest levels (Figure 8a). The largest deposition flux was observed on 25$^{th}$ June at the 24 m with 46 nmol m$^{-2}$ s$^{-1}$ level in agreement with a peak of evapotranspiration (Figure 8b).

Remarkably, the following two days LE was nearly 50% less and ozone fluxes too. In general LE and ozone fluxes seem to be correlated but there were some exceptions (e.g. on 3$^{rd}$ and 4$^{th}$ July). The smallest fluxes were observed on 6$^{th}$ July during the rainfall events, after which the ozone fluxes showed a remarkable increase (7$^{th}$ July) even if ozone concentrations were lower, corresponding to an increase of soil water content and evapotranspiration fluxes.

The diel average course of ozone fluxes (Figure 9) showed at all the levels the typical behaviour with low nighttime values

and the greatest deposition in the central hours of the day.

Ozone fluxes measured above the canopy (41 m and 32 m) showed very good agreement, nearly overlapped during the day. Both increased very rapidly in the morning and then stayed almost constant (between 8 and 10 nmol m$^{-2}$ s$^{-1}$) from 9:00 to 16:00, when they started to decrease. At 24 m, fluxes were not constant in the central part of the day and they were on average 40% greater than the above canopy levels with average peaks around 15 nmol m$^{-2}$ s$^{-1}$. From 9:00 to 16:00, air layers

above canopy including the top of the crown (from 24 m to the top of the tower), seemed decoupled from the air below: the above layers were in superadiabatic conditions with intense air mixing while the below canopy experienced a thermal inversion which gradually expands towards the top of the canopy and even above after 16:00.

Greater fluxes at 24 m might be caused by the location of these measurements which are just in the transient region between well mixed superadiabatic air and the below canopy thermal inversion.

### 3.5 NO and NO$_2$ fluxes and concentrations

NO and NO$_2$ concentrations along the tower profile (excluding near ground soil at 0.15 m) were relatively low with a maximum early in the morning respectively around 2 ppb for NO and around 14 ppb for NO$_2$. Both NO and NO$_2$ concentrations did not show great differences along the vertical profile (Figure 10a and Figure 10b). The greatest differences between the bottom and top level were only around 1 ppb, for both compounds, very early in the morning, between 4:00 and

30 9:00.

At soil level (0.15 m) the behaviour was completely different for both compounds. NO was always greater than the above levels (from 5 m to 41 m) with two peaks (Figure 10a): the first one at 6:00 around 15 ppb and the second one in the



afternoon around 17 (nearly 20 ppb). $NO_2$ at soil level was relatively constant ranging from 7 to 12 ppb g); even in this case two peaks were observed: at 6:00 (10 ppb) and around 17 (11 ppb).

NO and $NO_2$ fluxes at ground level were almost always mono-directional with NO emitted from soil and $NO_2$ deposited to the ground (Figure 10c). A significant change in the emission rate of NO and in the deposition of $NO_2$ was observed after the

rainfalls happened between 6[th] and 7[th] July (Figure 10d).

The average diel course of soil fluxes showed an almost constant emission of NO with two decreases: the first one around 6:00 and the second one at 17:00. These two decreases of the emissions were strictly linked with the stratification of the air above ground: an increase in the concentrations in a stratified environment led to a reduction of the concentration gradient between soil/litter and the atmosphere thus reducing the emission in turn. The average diel course of $NO_2$ deposition

followed nearly symmetrically the behaviour of the NO soil emission with a pronounced reduction of the deposition early in the morning and a less intense one in the afternoon. The simultaneous peaks of NO and $NO_2$ fluxes (Figure 10e and Figure 5) highlights a gas phase titration with an ozone reduction by NO.

At the top canopy the net exchange of NO with the above atmosphere it is very small except in the morning with a deposition peak between 6:00 and 11:00 which reached -15 µg N $m^{-2}$ $s^{-1}$. This NO deposition is in relation with the weak NO

gradient which developed above canopy after $NO_2$ photolysis and ended when $NO_2$ concentrations reached a minimum determined by the photolytic equilibrium of $NO_x$.

## 4 Discussion

Whilst turbulence and heat fluxes inside tree canopies had been studied more extensively, only few studies had attempted to partition ozone fluxes by means of flux measurements at different in-canopy heights (Dorsey et al., 2004; Launiainen et al.,

20   2013).

The evaluation of flux profiles relies on the constant flux hypothesis, one of the most fundamental theories of micrometeorology (e.g. Arya, 1989). In the case of the Bosco Fontana measurements, one would expect that the fluxes measured at 41 m and at 32 m should be almost equal and then the deposition flux should decrease, in absolute terms, at the lower levels due the presence of different in-canopy sinks for momentum and ozone (stomata and surfaces of the leaves,

branches and stems), and sinks and sources for heat. Consistent with this expectation, measured fluxes of heat and momentum were significantly reduced within the canopy, but fluxes at 32 m exceeded those measured at 41 m by on average 20%. This may be due to differences in the flux footprint, coupled with heterogeneity in the canopy (Dalponte et al., 2007; Acton et al., 2016), but it does not appear to be dependent on wind direction. Because the 32 m measurements were made lower within the surface roughness layer, it is possible that fluxes were locally somewhat enhanced. Fluxes of ozone were

similar at these two above-canopy heights, within the uncertainty of the measurement, but unlike for the fluxes of heat and momentum, measurements at 24 m showed a significantly larger ozone deposition than the two upper levels (32 m and 41 m) at certain times of the day. In the morning (9:00 to 12:00) 24 m ozone fluxes were on average nearly 3 nmol $m^{-2}$ $s^{-1}$ larger



than above the canopy (**Figure 9**), while they were nearly equal on average from 13:00 to 18:00 (fluxes at 24 m were only 0.5 nmol $m^{-2}$ $s^{-1}$ larger).

In order to investigate alternative reasons behind the enhancement of the ozone fluxes at 24 m in the morning and, at the same time, the higher sensible heat fluxes at 32 m, a spectral analysis was performed to compare the normalized cospectra of

the ozone fluxes and of the sensible heat fluxes at the different levels. Figure 11 shows the average normalized cospectra of the vertical component of the wind and ozone (on the left, Figure 11 a and c) and of w and temperature (on the right, Figure 11 b and d), the first two graphs referred to the measurements performed at 11:00 (i.e. during the morning ozone enhancement at 24 m) while the others referred to the measurements at 14:00 (i.e. when the 24 m ozone fluxes are comparable with the upper ones).

The average sensible heat cospectra at 32 and 41 m showed a very similar behaviour for 11:00  (Figure 11b), relatively noisy, with almost all peaks at the same frequency and with similar spectral intensity; only two peaks (0.003 Hz and 0.029 Hz) were observed at one level and not in the other one. At 15.00 both cospectra had a much irregular behaviour in particular the contribution of frequencies ranging from 0.02 Hz to 0.2 Hz was greater at 32 m than at 41 m (Figure 11d) but no additional indication was found in the spectral analysis to justify the higher sensible heat fluxes at 32 m.

By contrast, a qualitative explanation of this process can be found investigating the raw data. **Figure 12** illustrates a situation which was observed in several half hourly periods, showing the fluctuations around the average of the temperatures recorded at 24, 32 and 41 m. At 24 m the temperature signal due to the canopy heating which, in turn, heated the surrounding air, led to the formation of thermal bubbles and ramp structures. These thermal bubbles rose from the canopy and the increases of temperature observed at 32 m were much larger than those at 41 m, thus explaining the greater sensible heat fluxes at 32 m.

Because such convective cells are local phenomena, the flux at 32 m may on average have been more locally affected by the phenomenon than the flux taken at a higher height, which would integrate better over a larger area.

The cospectra analysis performed for the ozone fluxes did not provide an obvious explanation for the enhancement of the fluxes observed at 24 m in the morning; apart from the ozone cospectra at 16 m which had a very irregular behaviour, the three other cospectra did not show any particular difference which could be able to explain the higher ozone fluxes at 24 m:

the observed decrease of the ozone cospectra at 24 m and lower for frequency above 0.1 Hz is consistent with the notion that within the canopy the mean eddy-size is dictated by the canopy height.

The ozone chemical sink was identified as the responsible process for the enhanced flux at 24 m; this sink was due to the convergence of two NO fluxes at the top of the canopy: the NO deposition flux from the air above the forest and to the NO emission flux uprising from the forest floor.

The role of the NO related $O_3$ sink at the top canopy can be argued by considering the differences between the $O_3$ fluxes observed at 24 m and those at 32 m. In fact, the integration of these differences from 6.00 to 12.00 in the morning gives a value of 59.4 $\mu$mol $m^{-2}$ which is almost equal to the amount of NO converging to the top canopy level both from above and from below canopy, which integrated on the same hours gives 54 $\mu$mol $m^{-2}$.



This fact can be seen in Figure 13 where, supposing a stoichiometric reaction between NO and $O_3$ at canopy level, a modified graph for the fluxes at 24 m was introduced by subtracting the NO fluxes both from above and below canopy to the original fluxes at 24 m, leading to a much better agreement with the 32 m ozone fluxes.

The contribution of the NO emitted by soil to this ozone sink was relatively constant and greater than the contribution of the NO deposited from the atmosphere above, because this latter was very low in the afternoon and relatively high only in the morning (Figure 10e). This observation can be connected to the fact that the top canopy enhancement of the ozone sink was observed only in the morning and ceased in the afternoon.

A possible explanation of that could be found in the forest-atmosphere decoupling during the afternoon compared to the forest-atmosphere coupling observed in the morning. In fact, the increase of the NO concentrations at soil level (0.15 m,
Figure 10b) after midday, followed by the decrease of $O_3$ concentrations at the same level (Figure 5), suggests an air stratification inside the canopy and its decoupling from the above canopy air, as also found by Rummel et al. (2002) and Foken (2008). On the contrary the decrease of the soil level NO concentration (Figure 10b) from 6:00 to 12:00 in the morning suggests a relatively well-mixed canopy which is better coupled with the atmosphere above. This condition allowed also ozone and NO from the above canopy air to penetrate more easily into the canopy (see Figure 6b and the morning peak
of Figure 10b).

The afternoon stratification is also supported by the stability classes reported in Figure 4c and Figure 4d which were almost always stable or very stable both at 16 m and at 24 m from 15:00 to 18:00. In addition, the thermal inversion layer within the canopy increased its thickness during the afternoon (Figure 2b) rising form the 16 m observed around 12:00 to the 24 m observed from 14:00 to 16:00.

Again, the morning coupling and the afternoon decoupling is supported by the diurnal course of the specific humidity observed below canopy (Figure 1c). In the morning the almost constant amount of water vapour above and below canopy (a part from the top-canopy 24 m level where there was an unidentified process removing water vapour) reveals an efficient mixing of the air while in the afternoon the increase of the specific humidity from the three lower levels, due to soil evaporation and understorey transpiration, reveals an air stratification below canopy and a forest decoupling from the above
atmosphere.

The availability of ozone flux measurements at different heights within and above the canopy allowed a partition of the ozone fluxes among the different ecosystem layers: upper and lower crown, understory and soil. To do that we have assumed the ozone flux measured at 32 m as a total deposition flux, then we calculated the NO sink as the sum of NO deposited from the top and emitted from soil assuming a stoichiometric reaction between NO and $O_3$. The ozone taken up by the
upper/dominant crown was identified as the difference between the ozone fluxes measured at 32 m and those measured at 16 m (ignoring the apparently enhanced values at 24 m), while the ozone taken up by the dominated/lower crown was obtained as the difference between the ozone fluxes measured at 16 m and 5 m. Finally the deposition to the forest floor (soil and the understory vegetation) was calculated, as a residual of the difference between the ozone deposition at 5 m and the NO emitted from soil.





The result of this exercise is shown in Figure 14, where it can be observed that the dominant layer of the forest removed about 1/3 of the total deposited ozone while the dominated layer of the forest removed the main part of the ozone (46.5%). The whole forest canopy removed nearly 80% of the ozone deposited to the ecosystem, but it is worth noticing that this amount includes both stomatal and non-stomatal uptake.

A small midday depression of the deposition ozone fluxes can be observed in the deposition to the dominant crown while the diel course of the deposition to the dominated layer was influenced by the irregular behaviour of the 16 m ozone fluxes.

Only a minor part of $O_3$ was removed by the understory vegetation or deposited to the soil (2.0%) while an important role was played by the NO sink, mainly due to soil emissions, which accounted for the 18.2% of the total ozone deposition.

This latter result is in general agreement with the observations of Dorsey et al. (2004) who found that in a Douglas fir

plantation between 7% and 14% of the ozone deposition in the daylight hours could be attributed to the reaction with the soil emitted NO, while this fraction increased up to 41% during the night. Similarly, Pilegaard (2001) found that the NO sink accounted for a 25% of the ozone deposition in a Norway spruce forest, with an increase of this fraction up to 31% during the night.

Nearly all the nighttime ozone deposition at Bosco Fontana can be attributed to the NO depletion, which resulted the

responsible process for the observed ozone deposition at night. The fact that NO reaction accounts for 100% of the nocturnal $O_3$ sink would imply that other non-stomatal sinks are negligible during that time. It cannot completely ruled out, however, that our stirred soil flux chambers could somewhat overestimate the nocturnal soil NO emission, due to the enhanced amount of mixing in the flux chamber compared with the true forest floor during calm nights. Similarly, the analysis assumes that the only sink of NO is reaction with $O_3$. A small amount of uptake of NO by vegetation is possible. Overall this ecosystem did

not behave as a net NO emitter because the whole NO produced at soil level is consumed within the canopy, but as a weak NO sink because of the small amount of NO is received from the atmosphere in the first hours of the morning (Figure 10d). This differs from the observation of Dorsey et al. (2004) who estimated that nearly 60% of the NO emitted from the soil of a Douglas Fir forest escaped the trunk space to react aloft.

### 5 Conclusions

Ozone flux measurements were run along a vertical profile with five measuring point above, inside and below the canopy. The ozone flux measurements of the higher levels above the canopy were in good agreement between them and comparable with measurements on different forest ecosystems found in literature, where no measurements on oak-hornbeam forests like Bosco Fontana were found. Ozone fluxes at 16 m and 5 m showed a significant reduction of the ozone deposition measured above the canopy, while at canopy height (24 m) fluxes were higher than the above ones with a significant enhancement in

the morning. The main cause of this enhancement has been attributed to an ozone sink due to a reaction with NO both emitted from soil and deposited from the atmosphere.



Ozone flux dynamics observed at Bosco Fontana were complex: in particular the morning enhancement of the ozone fluxes at 24 seems to be favoured by the coupling between the forest and the atmosphere while in the afternoon the decoupling and the in-canopy stratification led to 24 m ozone fluxes comparable to the above canopy ones.

Most of the ozone, nearly 80%, was removed by the forest canopy: in particular the dominant layer removed 33.3% of the
5    ozone deposited while the dominated layer 46.3%. Only a little part was deposited on the soil or understory (2%), while the remaining part (18.2%) was removed by a chemical ozone sink due to the reaction with NO.

Finally the collected data can be useful for the parameterization and the fine tuning of process models whose aim is to better investigate the intra-canopy dynamics.

10    *Acknowledgements.* The author are in debt with the administration and the personnel of the Bosco Fontana National Reserve for their availability and continuous support. They are also grateful to the European Union for having fund the project ECLAIRE under which this campaign was made possible.

Finally, this publication was been funded by the Catholic University of the Sacred Heart in the frame of its Programs of promotion and dissemination of the scientific research.



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





**Table 1. Instrumentation installed at each height on the tower and on the nearby mast**

| Level / Height (m) | Ultrasonic anemometer | Fast ozone analyzer | Other fast analyzer | Slow sensors | |
|---|---|---|---|---|---|
| **41** | USA1 *(Metek, D)* | COFA *(Ecometrics, I)* | LI-COR 7500 *(Li-Cor, USA)* | HMP45<br>NR-lite<br>BF5<br>PTB101B<br>Rain gauge 52202 | *(Vaisala, FIN)*<br>*(Kipp & Zonen, NL)*<br>*(Delta-T Devices, UK)*<br>*(Vaisala, FIN)*<br>*(Campbell Scientific, USA)* |
| **32** | HS50 *(Gill, UK)* | ROFI *(CEH, UK)* | CLD780TR *(Ecophysics, CH)* | HMP45 | *(Vaisala, FIN)* |
| **24** | Windmaster PRO *(Gill, UK)* | FROM *(NOAA, USA)* | - | HMP45 | *(Vaisala, FIN)* |
| **16** | Windmaster PRO *(Gill, UK)* | COFA *(Ecometrics, I)* | - | HMP45 | *(Vaisala, FIN)* |
| **11** | - | - | - | HMP45 | *(Vaisala, FIN)* |
| **5** | R2 *(Gill, UK)* | FOS *(Sextant, NZ)* | LI-COR 7500 *(Li-Cor, USA)* | - | |
| **1.5** | - | - | - | PT100 | *(Campbell Scientific, USA)* |
| **0.15** | - | - | - | PT100 | *(Campbell Scientific, USA)* |
| **Soil** | - | - | - | TDR mod 616<br>HFP01SC<br>PT100<br>Soil dynamic chamber system | *(Campbell Scientific, USA)*<br>*(Hukseflux, NL)*<br>*(GMR Strumenti, I)*<br>*(IMK-IFU, D)* |




a)

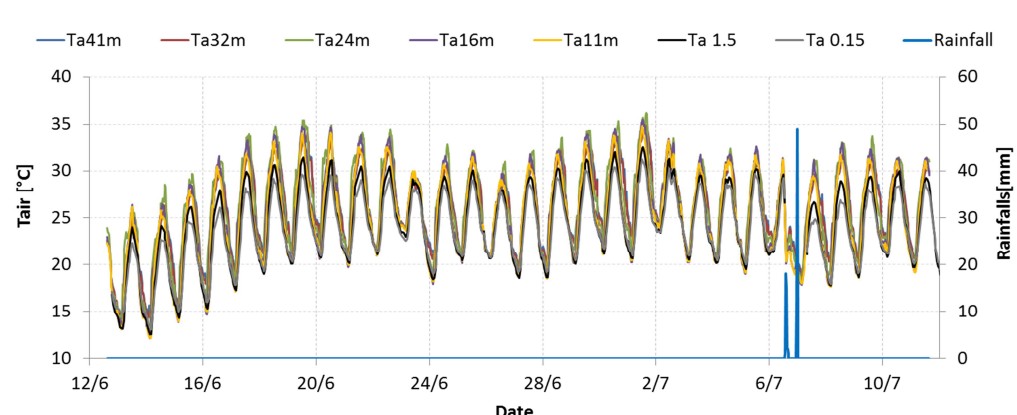

b)

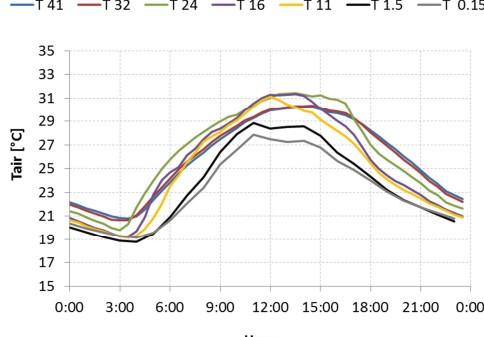

c)

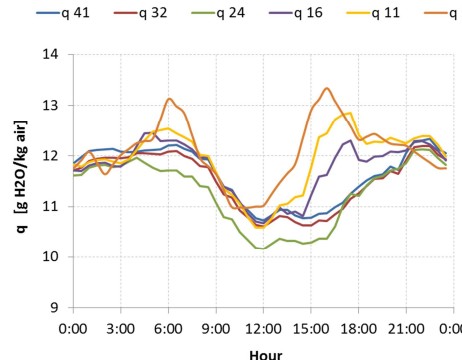

d)

e)





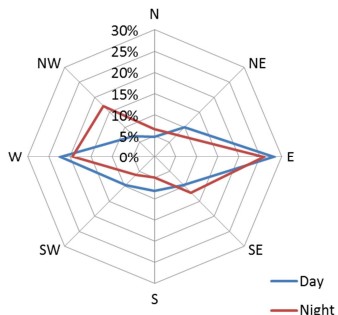

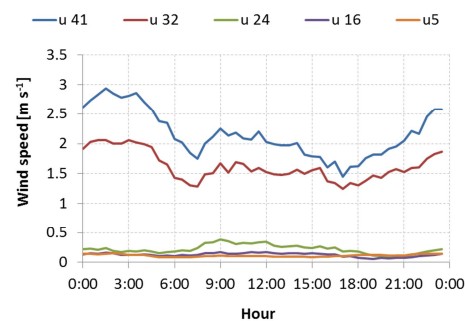

f)

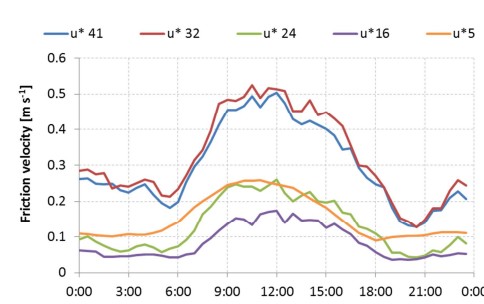

**Figure 1. a) Rainfall amounts and temperature evolution at the seven heights. Blue lines are rainfalls. b) Average diel course of air temperature at the seven heights. c) Average diel course of specific humidity at the five heights d) Wind rose based on 41 m data, the radial axis unit indicates the percentage of the data in each direction, the blue line diurnal data, the red line nighttime data. e) Average diel course of wind intensity at the five heights f) Average diel course of friction velocity at the five heights. For figures a), b), c) e), and f), the numbers in the curves label of the legend represent the measurement height (in meters).**



a)                                                    b)

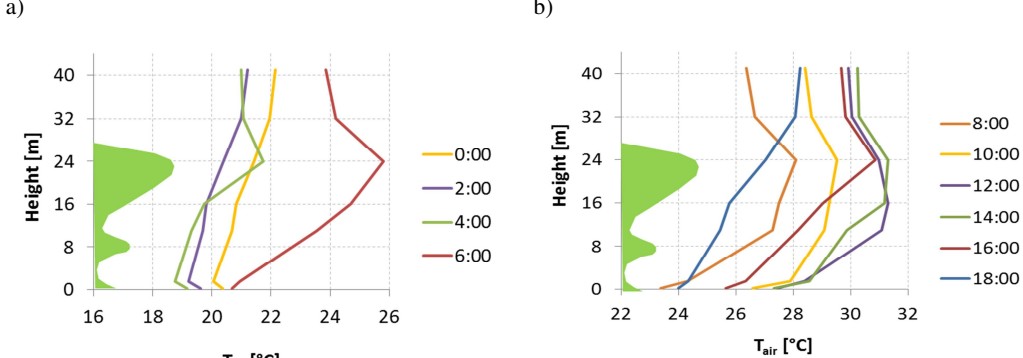

**Figure 2. Diurnal evolution of vertical profile of air temperature. a) from 0:00 to 6:00 AM; b) from 8:00 to 18:00. The green shaded area represents the vertical distribution of vegetation.**





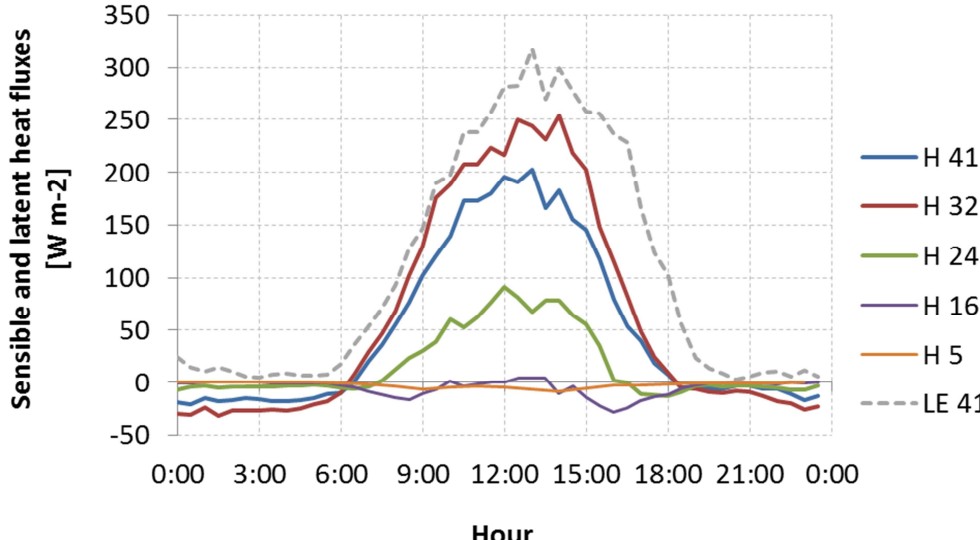

**Figure 3. Average diel course of sensible heat fluxes at the five levels (41 m, 32 m, 24 m, 16 m and 5 m, thick lines) and latent heat flux measured at 41 m (dashed line).**





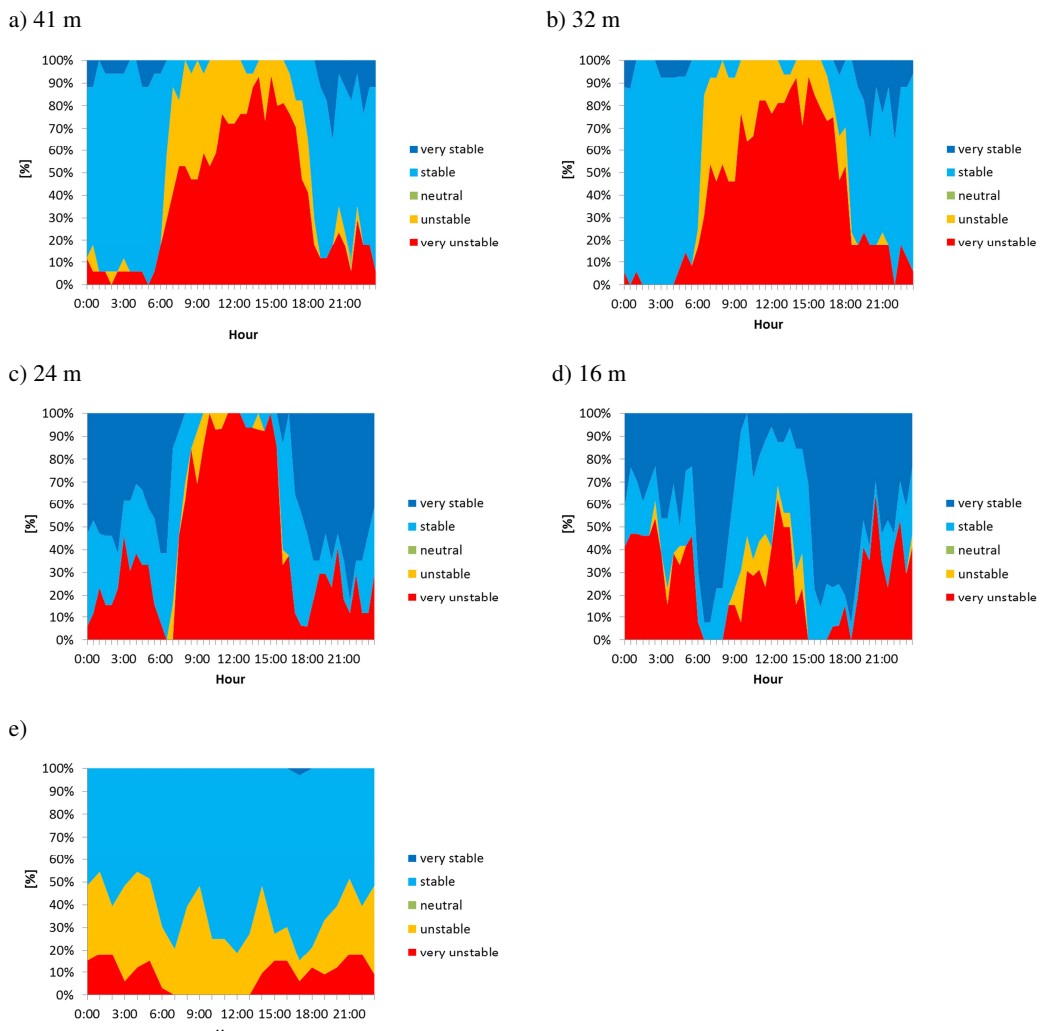

**Figure 4. Stability class distributions in the different hours of the day expressed as function of z/L for the different levels: a) 41 m, b) 32 m, c) 24 m and d) 16 m e) 5 m.**

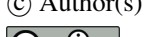



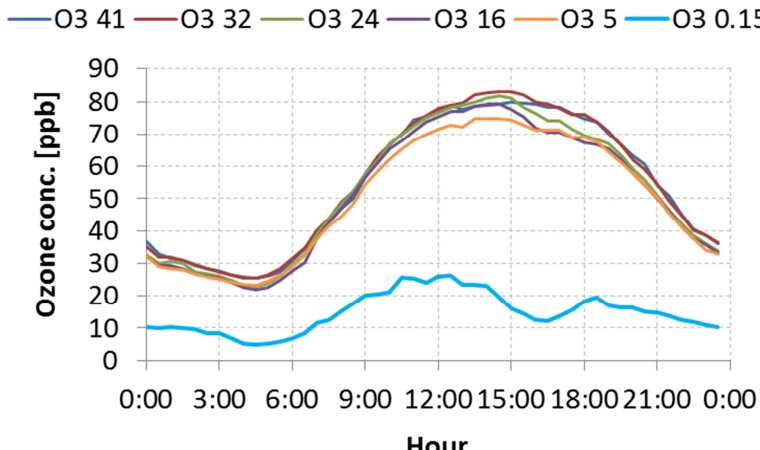

**Figure 5. Average diel courses of ozone concentrations at the six levels (41 m, 32 m, 24 m, 16 m, 5 m and 0.15 m).**



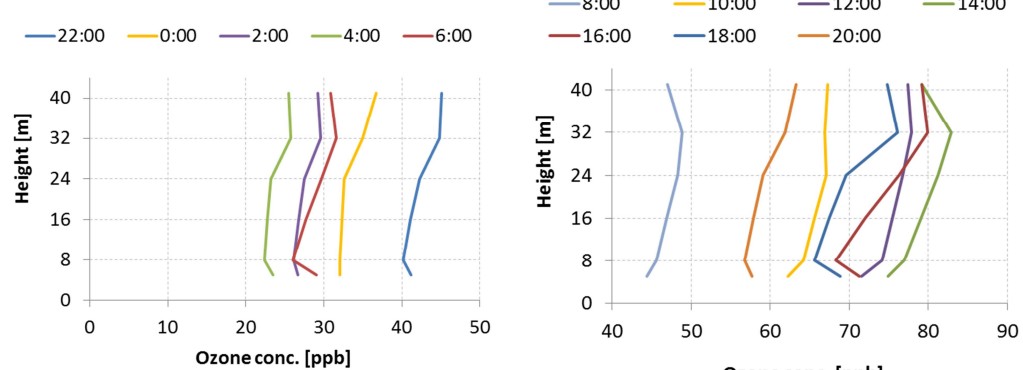

**Figure 6. Diurnal evolution of ozone concentration profiles: a) from 22:00 to 6:00; b) from 8:00 to 20:00. The time of the day to which measurements are referred is indicated in each figure label. The 0.15 m level has not been included here for a better visualization (ozone concentration at this level was around 10 ppb in a) and below 25 ppb in b)).**



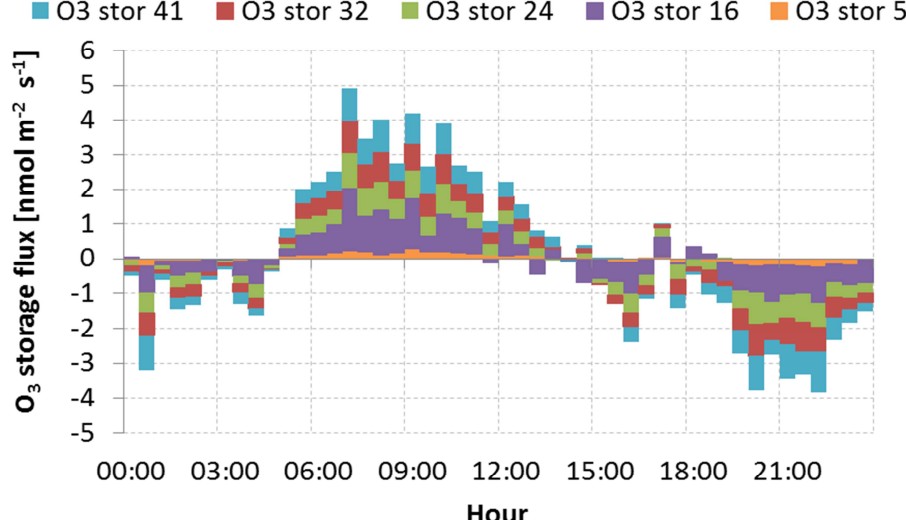

**Figure 7. Mean diel evolution of the ozone storage flux (the storage term of the Eq. 3). The contribution of the air column between adjacent flux measurement levels is indicated with a different color.**



a)

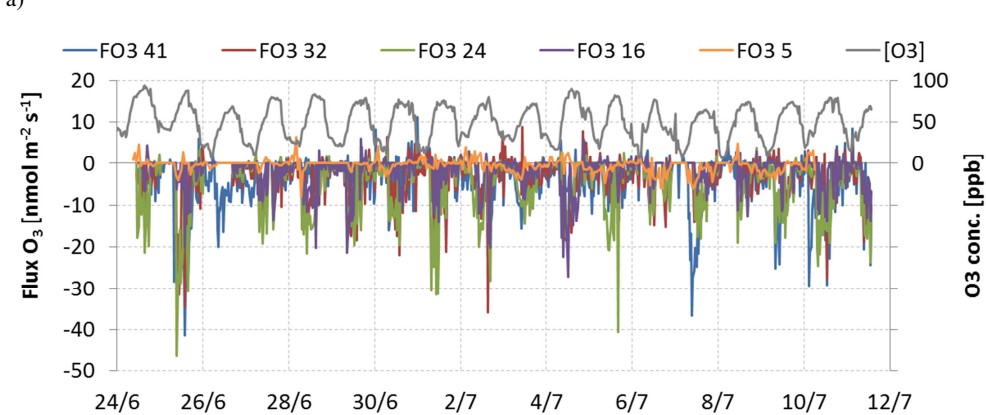

b)

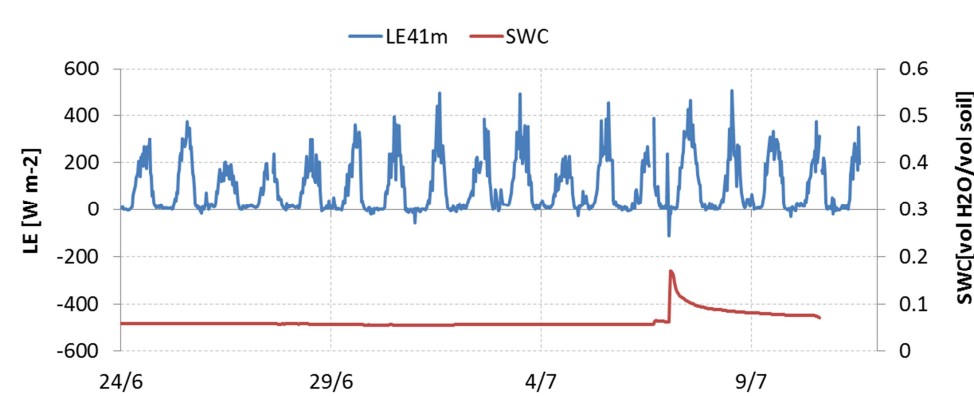

**Figure 8. a) Ozone fluxes at the five levels during the profile period (41 m, 32 m, 24 m, 16 m and 5 m). b) Latent heat fluxes measured at the top of the tower and soil water content expressed as volumetric ratio between water and soil**





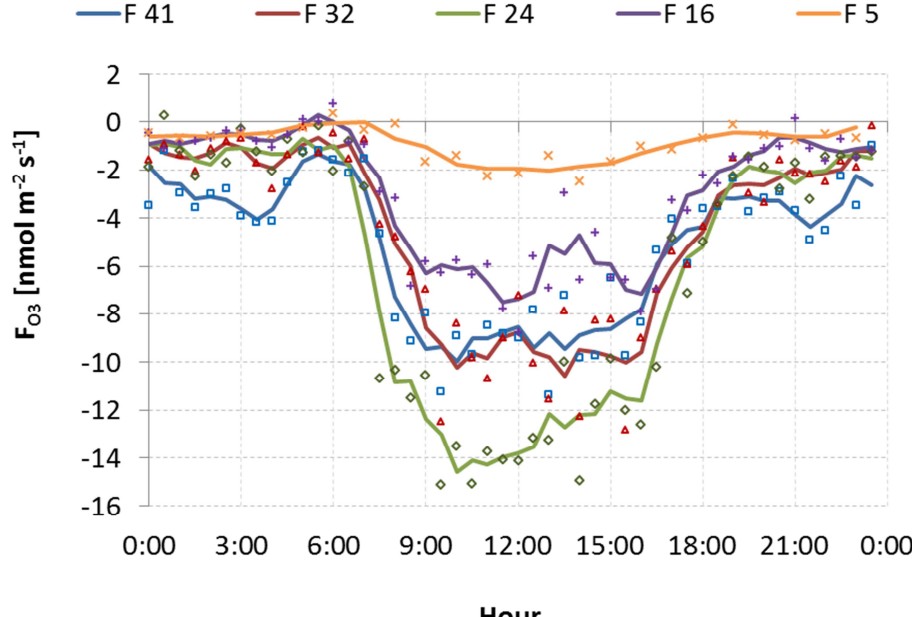

**Figure 9. Average diel courses of ozone fluxes during the profile period at the five levels (41 m, 32 m, 24 m, 16 m and 5 m), dots represent half an hour averages while lines are one hour and half running means centered on each half an hour.**



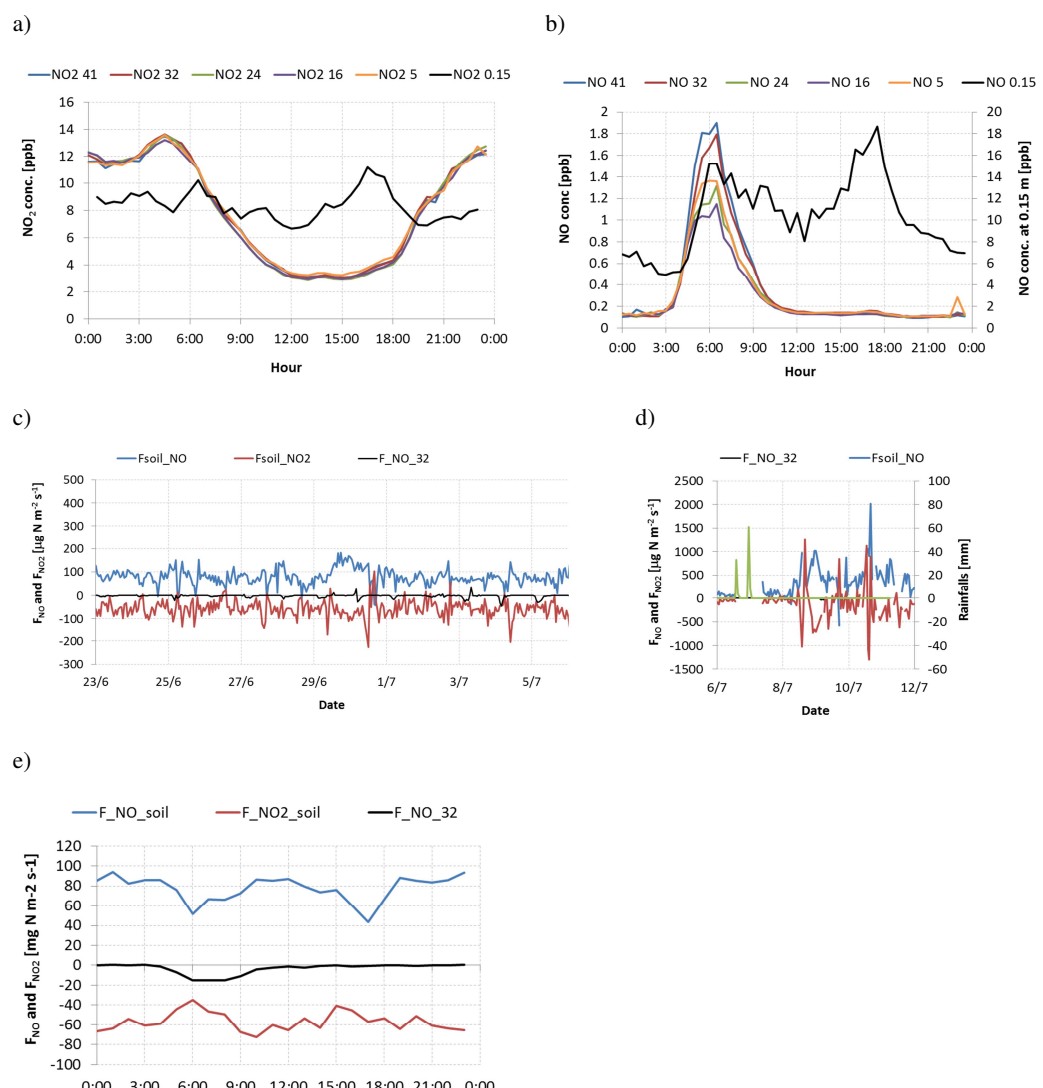

**Figure 10. NO and NO₂ concentrations and fluxes during the profile period. a) Average diel course of NO concentrations at the five levels (41 m, 32 m, 24 m, 16 m and 5 m); b) Average diel course of NO₂ concentrations; c) Soil NO and NO₂ fluxes and NO fluxes at 32 m before rainfalls events; d) Soil NO and NO₂ fluxes and NO fluxes at 32**



**m after rainfalls events. Please note the different scale between c) and d); e) Average diel course of soil NO and NO$_2$ fluxes and of NO fluxes at 32 m.**



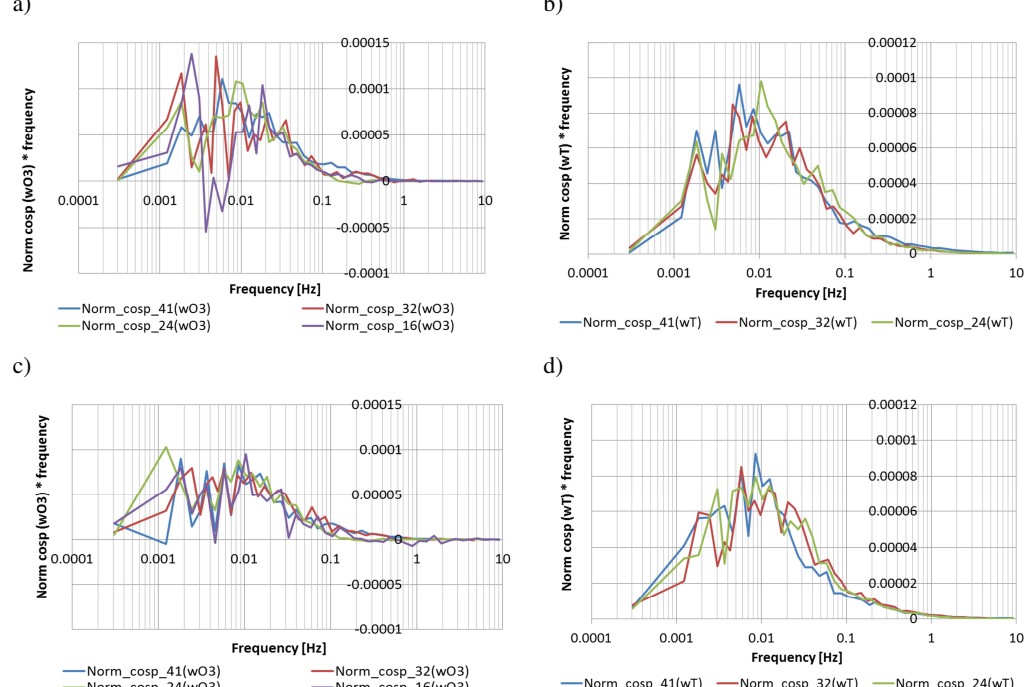

**Figure 11 Average normalized cospectra of the vertical component of the wind and ozone (on the left) and temperature (on the right). The first row is referred to the average of all the available measurements at 11.00 while the second row to all the measurements at 15.00.**


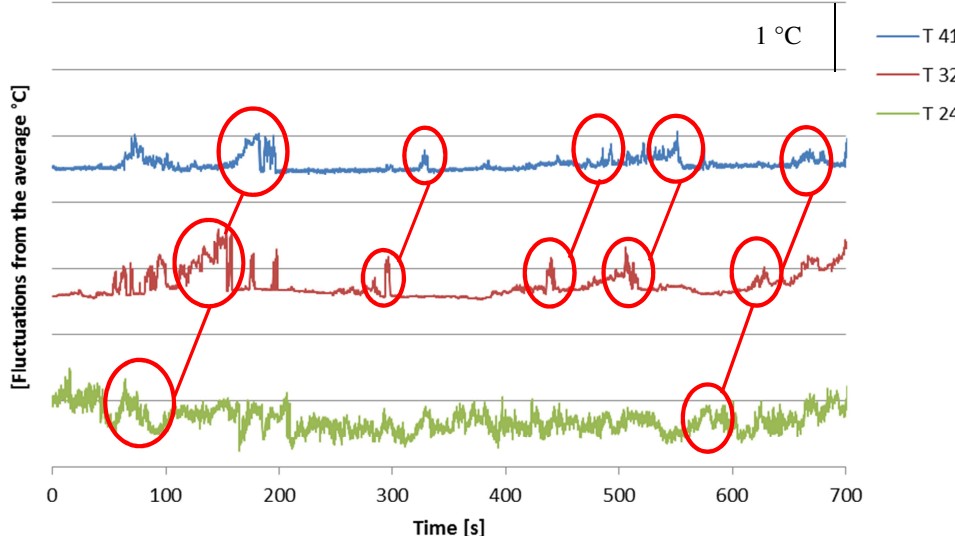

**Figure 12** Fluctuations of temperature around their average values measured on 5[th] July, 13:00. Blue line is referred to measurements at 41 m, red line to 32 m and green line to 24 m.



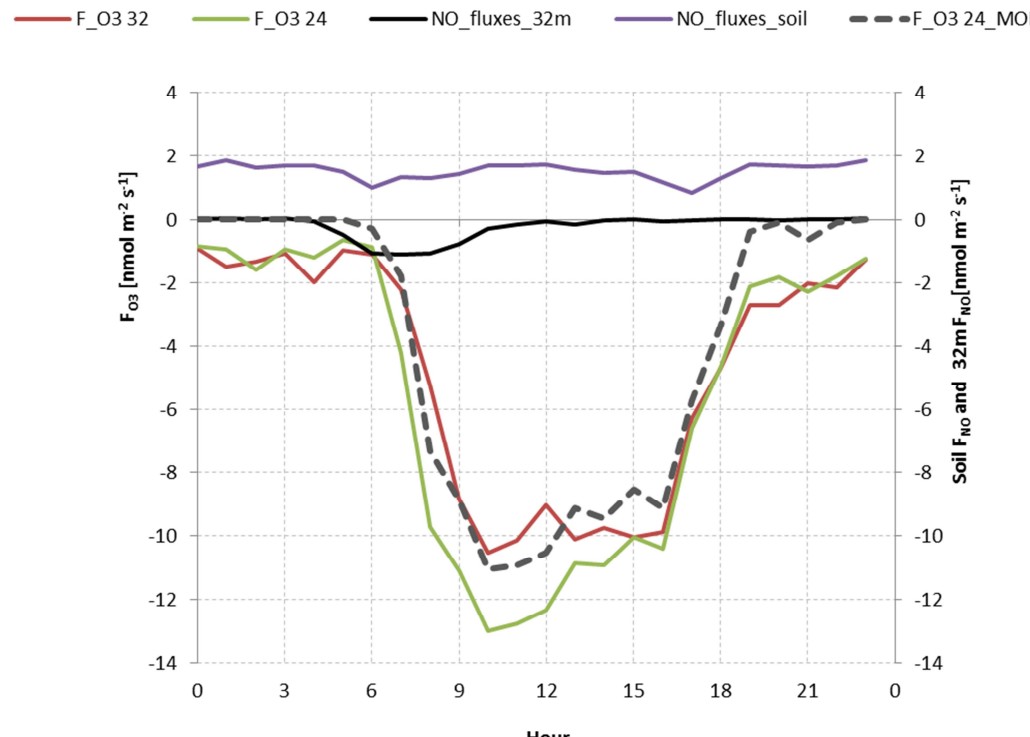

**Figure 13** Average diel course of ozone fluxes at 32 m (red line), ozone fluxes at 24 m (green line), NO fluxes at 32 m (black line) soil NO fluxes (purple line) and modified ozone fluxes at 24 m (dashed grey line). This latter takes into account the role of the NO sink.



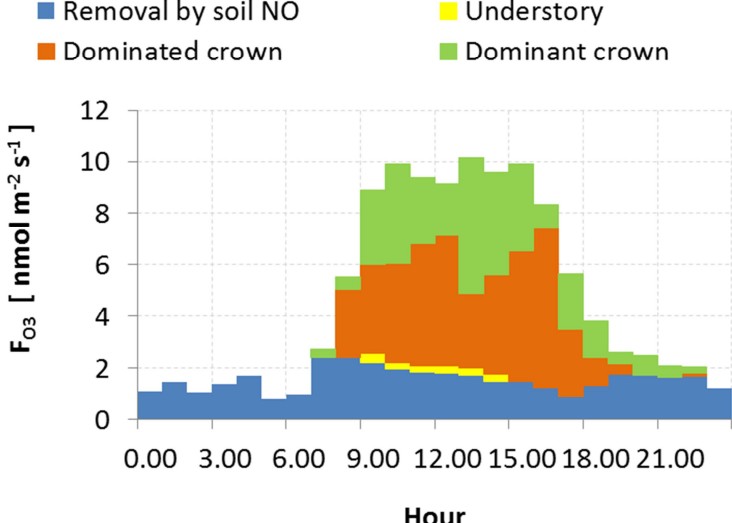

**Figure 14 Average diel course of the flux partition among dominant/higher crown (green area), dominated/lower crown (orange area), soil and understory (yellow area) and NO sink (blue area).**