# Peer review of "Characterization of ozone deposition to a mixed oak-hornbeam forest. Flux measurements at five levels above and inside the canopy and their interactions with nitric oxide"

_Atmospheric Chemistry and Physics, 2018_

## Referee Comment (RC1) · Anonymous Referee #1 · 22 May 2018

Finco et al. (2018) present a novel dataset of ozone and NOx fluxes at multiple levels from a month-long field campaign at a forest in Italy. The authors find that NO plays an important role on the observed ozone fluxes; ozone and NO reaction dominates the observed ozone flux near the canopy floor and determines the differences in observed ozone fluxes at higher levels in the canopy. The authors find that about half the ozone flux during the day actually occurs to the middle of the canopy, which is very interesting; it's typically considered that most of the ozone is deposited to the upper third of the canopy. At night, reaction of ozone and NO close to the ground dominates

the ozone flux. The weakness of the paper is that there is quite a lot of data presented (thirteen figures) but the direction of the manuscript and the significance of the authors' findings are not always clear. I would encourage the authors to better guide the reader and contextualize their findings, clearly articulating how their study advances current understanding. One way of better guiding the reader would be to hone the paper's objectives and including some overarching statements at the beginning of each section and move some of the figures to supplementary material. After substantial revisions, I think this paper should be published in ACP.

General comments.

In the introduction, the authors motivate their work with ozone damage to ecosystems. But they do not introduce stomata, non-stomatal deposition pathways, or that air chemistry can impact observed fluxes. In the abstract, the authors mention that NOx can lead to ozone production, but in the introduction the authors mostly discuss ozone destruction by NO. Spelling out the connections and expanding their introduction would help readers follow along and see the value of the authors' approach.

One of the authors' stated objectives is "to test the capacity of existing deposition models to predict intra-canopy dynamics involving ozone reactions with NOx and VOCs". But I do not see any analysis in the paper addressing this objective.

Why aren't there confidence intervals on the figures with averages?

The current contents of the discussion seem like they would better fit in the results, as there is a lot of new data analysis.

There are many figures, and relatively little text. My feeling is that many of the figures could be moved to supplementary material.

It would be helpful to the reader if the authors were more generous in referencing their figures. For example, on page 11, lines 16-24 there is no reference to any figures.

The authors should check their in-line references for commas after the author name or

"et al." and whether their use of commas or semi-colons between references is consistent (and ACP policy as to whether there is a difference between in-line references used with "e.g." vs. not).

In general, I would recommend that the authors do not use words that express a value-judgement, like "extremely", "remarkable", "peculiarity".

Line-by-line comments.

Abstract.

Generally readers are not going to know what "in the framework of the European FP7 project ECLAIRE" means.

Instead of saying "A partition of the ozone fluxes will be shown to identify the most relevant sinks" the authors should say report their most important finding in their abstract.

Page 2, Line 4: "had" should be "has"

Page 2, Line 30-31: It's not clear to me why this is relevant

Page 3, Line 4: What is a "climax" ecosystem?

Page 3, Lines 8 & 12: Here and in the remainder of the paper, I find it difficult to distinguish between the terms "dominant" and "dominated". Are there other ways the author could describe them?

Page 3, Line 13: Atmospheric chemists may not know what nemoral means.

Page 3, Line 24: How do the authors calculate the fetch size?

Page 3, Line 30: Will the authors please use "instrument models" or "instrument types" here instead of "model"?

Page 3, Line 31: 10 m away from the tower in the vertical? Or in the horizontal?

Page 4, Lines 4-10: Are COFA, ROFI, FROM, and GSAS acronyms? If so, it would be

helpful to spell out what they stand for.

Page 4, Line 19: Why how the data are stored (i.e., in hourly files) relevant?

Page 4, Line 26-27: I'm not sure that I understand. Why is the chamber is only measured for six minutes? What do the authors use the measurement of the reference chamber for?

Page 5, Line 8: I don't think model should be abbreviated, here and elsewhere

Page 5, Line 22-24: The authors should check the consistency of their abbreviations. For example, Campbell Scientific is abbreviated in one sentence but in the next sentence it is not.

Page 6, Line 7-9: What was the result of the inter-comparison between the two COFAs?

Page 6, Line 10: Are the authors examining the relative standard deviation for a given half-hour?

Page 6, Line 11: Why is systematic in parenthesis?

Page 6, Line 22: I don't think including this sentence is necessary.

Page 7, Line 9: "were" should be "was"

Page 7, Line 10: What does "semi-hourly" mean?

Page 7, Line 13: "k" should be "K" in kinematic

Page 7, Line 26: Is V0 the initial voltage?

Page 7, Line 32: The "C" of "C_O3" is missing

Page 7, Line 33: "let" should be "allow"

Page 8, Line 17: What is "it" here?

Page 8, Line 7-19: What is the timescale that the authors correct for storage over?
Page 8, Line 24: There is a "the" missing from between "cooled" and "air"

Page 8, Line 28: Is this local time? If so, it would be more straightforward to refer to it as local time.

Page 8, Line 29: Do the authors mean the respective lower and higher levels by "other ones"?

Page 8, Line 29-30: I think the authors mean that only two significant rainfalls occurred during the campaign and they happened at the end of the campaign

Page 11, Line 10: Why is this remarkable?

Page 11, Line 11: Why would ozone fluxes and LE fluxes be correlated?

Page 12, Line 1: There is a rogue "g);" in this line that needs to be deleted

Page 12, Lines 7-8: But there weren't necessarily decreases in emission, there were just decreases in the observed flux (perhaps one could call this the effective emission?)

Page 12, Lines 9-10: Instead of "followed nearly symmetrical" I would say they were close to inversely proportional

Page 12, Lines 11-12: Do the authors mean that the simultaneous minimums?

Page 12, Lines 24-16: The authors should refer to the figures that they are concluding this from. Will the authors please be more explicit about what they mean by "in relation to" here? Why is NO deposition most apparent at this time?

Page 12, Lines 22-25: Are the authors talking about the decreasing instrument footprint size at lower levels of the canopy? I think their logic needs to be spelled out a little more.

Page 13, Line 12: The authors should be careful as to whether they refer to time as HH.MM or HH:MM, here and elsewhere

Page 13, Lines 4-21: I find the discussion of the sensible heat fluxes interesting, but tangential. It would be helpful to the reader if the authors did not digress.

Page 13, Lines 27-34 & Page 14, Lines 1-7: This text should be one paragraph, not three. Also, I would avoid using the word "fact"

Page 14: Lines 1-3: I am not sure I am following this description well. It would be helpful if the authors more clearly spelled out their calculation. Perhaps pointing the readers to compare the grey dashed line and the red line would be helpful for readers.

Page 14, Line 4: instead of "to this ozone sink", will the authors say something like "to the observed ozone flux"?

Page 14, Lines 4-5: By "the contribution of the NO deposited from the atmosphere above", are the authors talking about the contribution of NO transported into the forest to the observed ozone fluxes?

Page 14, Line 6: What is the "top canopy enhancement of the ozone sink"?

Page 14, Line 8: The "of that" is unnecessary

Page 14, Lines 8-25: This should be one paragraph, not three

Page 14, Lines 33-34: Please clarify the last calculation - "the residual of the difference of ozone deposition at 5 m and the NO emitted from soil". Also, do the authors perform this calculation on half-hourly data or on the campaign averages for a given hour?

Page 15, Line 3: Specify that it's ecosystem removal relative to air chemistry removal

Page 15, Line 5: Isn't the depression in the ozone deposition to the dominated crown?

Page 15, Line 14: Specify NO depletion of ozone

Page 15, Line 17: What do the authors mean by "stirred"?

Page 15, Line 19: Why is only a small amount of uptake of NO possible? What are the references for this?

Page 15, Line 20: What do the authors' findings mean for the canopy reduction factor for NOx?

Page 15, Line 27: Which studies are these values comparable to?

General comments on figures and tables.

Table 1: What does the text in the parentheses mean? What do these instruments measure? What height is the nearby mast?

Figure 5: What are the distributions over? All the days in the measurement campaign? What is z? What is L? What are the values of z/L used for each class? The height label is missing for plot e).

Figure 10d: Please indicate the green line is rainfall. Also, is rainfall in the context of NO fluxes discussed?

Are all figures averages over the entire campaign? If so, please specify in the figure captions.

---

## Author Comment (AC1) · 23 May 2018

Many thanks to the first referee for his/her prompt review of our manuscript. We have appreciated the referee's comments about the contextualization of our result and the suggestion of driving the reader to the paper objectives in a clearer way. We will start working on the minor requests while we are waiting for the comments of the other referees. Then we will proceed to the extensve revision of the manuscript accordingly.

[Figure]

2018.

---

## Referee Comment (RC2) · Anonymous Referee #2 · 15 Jun 2018

Close examinations of the pathways controlling ozone deposition in the forest setting are important for understanding the oxidation chemistry in the forest, secondary organic aerosol formation, the boundary layer ozone budget, and, as mentioned in this manuscript, the impact of ozone on plant health. In this manuscript, the authors presented the ozone and NOx concentration gradient and flux data, and the associated meteorological parameters from the ECLAIRE campaign in 2012, as well as the initial data analyses, which would lead to a subsequent model analysis, as implied in the Introduction, that may generate model predictions of intra-canopy dynamics involving

ozone reactions with NOx and VOC. As a result of the month-long summertime observations of ozone and NOx at multiple height within and above a forest canopy at the most polluted area in Europe, this dataset provides a valuable case study of ozone dynamics and related canopy scale processes, and biosphere-atmosphere exchange. However, as specified below, major revisions are recommended.

The authors performed a fairly detailed treatment of the meteorological data to obtain the ozone fluxes at the measurement heights. Results of the fluxes at multiple elevations throughout the canopy provide information on the sources and sinks, thus the processes that affect the trace gas species in the forest. Given the data being from a month-long campaign under different meteorological conditions, the analysis could be strengthened and the conclusions may be better supported and possibly modified with the following additional considerations.

1) Above canopy influences from air transported to the surface layer above the canopy. According to the data, about 50% of the wind is from either east or west with the rest from other directions. Are there any differences in the quantities measured that coincide with the wind direction differences? Are the possible influence of the different amount of the pollutants (ozone and NOx) transported to the site considered?

2) The fluxes under the stable/unstable conditions within the canopy. If I understand it correctly, in Figure 4, y-axis is the percent of the measurement time. If so, there were times the entire canopy was either stable or unstable throughout a 24-hour period. It would be informative to separately analyze the data under these two regimes, especially when considering the within the canopy stability in the context of the enhanced ozone deposition flux at 24 m.

3) Dry and wet conditions. Apparently after the rainfall on July 6, the NO/NO2 fluxes increased dramatically. How did these changes affect the ozone deposition flux? In addition, it is known surface wetness affect ozone deposition as well. It would be instructive if the dry/wet conditions can be contrasted in the data analysis.

The data analysis results, as stated in the manuscript now, would be more convincing if the above-mentioned aspects were considered. One of the main conclusions in the manuscript is that the enhanced ozone flux at the canopy top is due to the combined NO fluxes from above the canopy and the soil emission, thus an enhanced chemical sink of ozone. However, there are other factors such as stomatal uptake and photochemical reactions involving NOx, O3 and BVOCs, both processes obviously associated with sunrise, that affect the ozone flux. The net result could well be an increased flux. Are there data available from this campaign that would help the authors address these possibilities?

It would help the readers to better understand and assess the data if the authors could present the time series plots, including error bars when appropriate, of the measurement results. It is also important to show the standard deviation (if the mean values are used) or the interquartile range (if median values are used) in the average diurnal course plots including Figure 6.

In Table 1 where it is not indicated or obvious, please list the measurements next to the instruments listed, for example, HMP45 (temperature, humidity).

I may have missed the point in Figure 8 but cannot readily see from which height are the plotted ozone mixing ratio results. Also if possible, please unify the tick location and tick labels in plots 8a and 8b.

I am not sure why the analysis of the enhanced O3 flux at 24 m is only for the morning (9:00-12:00). From the data, the enhancement, although decreasing after mid-day, lasts through 15:00. If, because of the scatter of the data, the fluxes were basically the same from 41 to 24 meters in the early afternoon, it needs to be shown and stated more clearly.

The authors used Figure 12 to explain the effect of the thermal bubbles and why the greater sensible heat flux at 32 m than at 41 m. However, the data shown are from 13:00 on July 5th. It is not clear whether this is a special case or an example of a

typical situation.

The manuscript could use some help with the English language usage and organization. For example, Page 2, line 4, "Prompted by its phytotoxicity" –> Because of the phytotoxicity of ozone, . . . Page 2, line 6, ". . ., 2013), thanks also to the. . .." may be changed to, for example: . . ., 2013). Eddy covariance measurements were made possible thanks to the . . . Page 2, line 16 – 18 and line 26-28, field campaign information was repeatedly given. Other editorial changes are not listed here since the reviewer has recommended major revisions.
* * *

---

## Short Comment (SC1) · 12 Jul 2018

Dear referee, we are finalizing the answers to your peer review and we would like to submit you a question about one of your request, you can find it in the attached file. best regards Giacomo Gerosa and co-authors

[Figure]

Dear referees,

we are finalizing the answers to your peer review and we would like to submit you a question about one of your request.

We understand the importance of the confidence intervals/error bars but we think that the readers might be confused because of too overlapping lines of the error bars and might not appreciate well the course of the plotted parameters. For this reason we have presented the graphs without error bars but if the referees persist we could add them in a following step, for instance in the next days when we will submit the final reviewed paper.

This is an example of the same figure with and without error bars and, in our humble opinion, we think the clearer figure is the one without error bars.

[Figure]

[Figure]

Another possibility we would like to propose to reviewers is it to add in the caption an indication of the range of the error bars, so that the reader can have an idea about them.

What do you prefer?

Best regards

Giacomo Gerosa and the co-authors

**Fig. 1.**

---

## Short Comment (SC3) · 12 Jul 2018

Dear referee we would like to draw your attention on another case (you can find it in the attached file) where error bars can be misleading. best regards

[Figure]

Dear referees,

we would like to draw your attention to another example, even more evident than the one that we have already published, where the error bars can be misleading of the diel average course of the plotted parameters. In this case the error bars hide the course of the ozone concentrations at the different heights when the lines are close. For example, it is nearly impossible to understand the course of the ozone concentrations at 41 m. So we would suggest to add some indicative values of the confidence intervals in the caption

[Figure]

*Figure 1 Average diel courses of ozone concentrations at the six levels (41 m, 32 m, 24 m, 16 m, 5 m and 0.15 m). The maximum and the minimum confidence interval were respectively ± 3.0 ppb and ± 1.7 ppb for 41 m, ± 3.0 ppb and ± 1.8 ppb for 32 m, ± 3.4 ppb and ± 1.9 ppb for 24 m, ± 3.4 ppb and ± ppb 1.8 for 16 m, ± 2.9 ppb and ± 1.7 ppb for 5 m and ± 3.4 ppb and ± 1.0 ppb for 0.15 m.*

**Fig. 1.**

---

## Author Response (AR1)

Finco et al. (2018) present a novel dataset of ozone and NOx fluxes at multiple levels from a month-long field campaign at a forest in Italy. The authors find that NO plays an important role on the observed ozone fluxes; ozone and NO reaction dominates the observed ozone flux near the canopy floor and determines the differences in observed ozone fluxes at higher levels in the canopy. The authors find that about half the ozone flux during the day actually occurs to the middle of the canopy, which is very interesting; it's typically considered that most of the ozone is deposited to the upper third of the canopy. At night, reaction of ozone and NO close to the ground dominates the ozone flux. The weakness of the paper is that there is quite a lot of data presented (thirteen figures) but the direction of the manuscript and the significance of the authors' findings are not always clear. I would encourage the authors to better guide the reader and contextualize their findings, clearly articulating how their study advances current understanding.
One way of better guiding the reader would be to hone the paper's objectives and including some overarching statements at the beginning of each section and move some of the figures to supplementary material.
After substantial revisions, I think this paper should be published in ACP.

RESPONSE
We thank the reviewer for the careful revision and his/her suggestions to improve this manuscript.
We have clarified the aims of the paper in order to better guide the reader as requested. For this sake the Introduction has been revised, enhanced and reshaped.
Some of the Figures and Tables have been moved to the Supplementary material, and the Discussion has been refined to make it more focused on the paper topics.

General comments.

In the introduction, the authors motivate their work with ozone damage to ecosystems. But they do not introduce stomata, non-stomatal deposition pathways, or that air chemistry can impact observed fluxes. In the abstract, the authors mention that NOx can lead to ozone production, but in the introduction the authors mostly discuss ozone destruction by NO. Spelling out the connections and expanding their introduction would help readers follow along and see the value of the authors' approach.

RESPONSE
Ok, the introduction has been improved and expanded according to the reviewer's suggestion by listing all the deposition pathways including stomatal deposition.
The connections between different parts are now improved.

One of the authors' stated objectives is "to test the capacity of existing deposition models to predict intra-canopy dynamics involving ozone reactions with NOx and VOCs".
But I do not see any analysis in the paper addressing this objective.

RESPONSE
The reviewer is right. We realized that this objective was improperly inserted among the objectives of this paper.
We think that data from this field campaign could be used to test the O3 deposition predicting capacity of some of the existing deposition models, particularly on the intra-canopy dynamics of O3 and NOx.
We are aware that a new model named ESX is under development as a follow up of the ECLAIRE project and that data presented in this paper will be used to calibrate and validate it. Our sentence was meant to be referred to this follow up.
Thus, the text has been modified as follows:
The detailed dataset of this campaign will also allow future tests on the capacity of existing deposition models to correctly predict ozone deposition dynamics on forest ecosystems, and particularly intra-canopy dynamics involving ozone reactions with NOx.

Why aren't there confidence intervals on the figures with averages?

RESPONSE
We have already posted two comments in the discussion about this referee's remark. In our humble opinion we think that in many figures of our manuscript, because of the presence of five lines (or more) in the figures, the use of error bars could be misleading for the reader because error bars are overlapping and could hide the diurnal course of the plotted parameters.
For example, please refer to the following graph reporting the diel course of [O3] at six levels. The error bars hide the course of the ozone concentrations at the different heights when the lines are close. For example, it is nearly impossible to understand the course of the ozone concentrations at 41 m.

[Figure]

However, we agree with the reviewer that the explication of the uncertainties of the means is of primary importance. So, we would suggest to report the range of the variation of the standard error for each curve (from the minimum to the maximum values observed in the 24h) in the text of the figure captions, without losing statistical robustness and the possibility to clearly appreciate the diel courses.
The following Figure reports and example of how we cope with this issue throughout the manuscript.

[Figure]

**Figure 1 - Average diel courses of ozone concentrations at the six levels (41 m, 32 m, 24 m, 16 m, 5 m and 0.15 m). The maximum and the minimum standard error of the half-hourly means were respectively 3.0 and 1.7 ppb for 41 m, 3.0 and 1.8 ppb for 32 m, 3.4 and 1.9 ppb for 24 m, 3.4 and 1.8 ppb for 16 m, 2.9 and 1.7 ppb for 5 m, and 3.4 and 1.0 ppb for 0.15 m.**

The current contents of the discussion seem like they would better fit in the results, as there is a lot of new data analysis.

RESPONSE
We understand the point of view of the reviewer. However, while the data analysis reported in the Results are merely descriptive and represent the groundwork for the following Discussion, the new data analyses shown in the Discussion are functional to support the interpretations of the Results. Moreover, the data analyses in the Discussion summarize several different results and help us to draw the Conclusions.
In our opinion, moving part of the Discussion in the Results would also implicate many cross-references to other results that could be still unexplained in that part of the manuscript.

There are many figures, and relatively little text. My feeling is that many of the figures could be moved to supplementary material.

RESPONSE
OK, we have moved to supplementary material some figures which are not essential for the interpretation of the behavior of the gas exchange at the forest (Table 1, Figure 1a, 1b, 1d, 1e, Figure 7, Figure 8b) and figures, like the cospectra ones (Figure 11a, 11b, 11c, 11d), which were used only for excluding some tangential hypotheses.
We have tried to keep in the manuscript only the most relevant figures.

It would be helpful to the reader if the authors were more generous in referencing their figures. For example, on page 11, lines 16-24 there is no reference to any figures.

RESPONSE
Ok, thank you. Some additional references to the figures have been added in the text, as suggested.

The authors should check their in-line references for commas after the author name or "et al." and whether their use of commas or semi-colons between references is consistent (and ACP policy as to whether there is a difference between in-line references used with "e.g." vs. not).

RESPONSE
Ok, corrected.

In general, I would recommend that the authors do not use words that express a value-judgement, like "extremely", "remarkable", "peculiarity".

RESPONSE
Ok. These adjectives have been removed throughout the manuscript.

Line-by-line comments.
Abstract.
Generally readers are not going to know what "in the framework of the European FP7 project ECLAIRE" means.

RESPONSE
The reference to the ECLAIRE project has been deleted from the abstract and moved to the acknowledgements.

Instead of saying "A partition of the ozone fluxes will be shown to identify the most relevant sinks" the authors should say report their most important finding in their abstract.

RESPONSE
Ok, thanks. The abstract has been modified accordingly to this suggestion.

Page 2, Line 4: "had" should be "has"

RESPONSE
Thank you for reporting it to us. However, in the revised version of the Introduction the whole sentence has been removed.

Page 2, Line 30-31: It's not clear to me why this is relevant

RESPONSE
In the revised version of the Introduction this sentence has been slightly modified and it was meant to underline that this paper is part of a group of companion papers which present different aspects of the same joint monitoring campaign performed under the ECLAIRE project. Since other reviewers asked for information about the interaction of O3 with VOCs, this sentence is simply functional to direct them to the papers which explain that in details.

Page 3, Line 4: What is a "climax" ecosystem?

RESPONSE
A climax ecosystem is an ecosystem at its maximum development stage (mature). At this stage the ecosystem is at a steady state from an energetic and ecological point of view.
We have added a short description of it in brackets in the text.

Page 3, Lines 8 & 12: Here and in the remainder of the paper, I find it difficult to distinguish between the terms "dominant" and "dominated". Are there other ways the author could describe them?

RESPONSE
These are typical ecological terms. The dominant tree layer is composed by the higher trees of the ecosystem, while the dominated layer is made up of the lower trees and the understorey. However, following the reviewer's suggestion, in order to help the reader unfamiliar with these ecological terms we adopted other more common terms to describe them.

Page 3, Line 13: Atmospheric chemists may not know what nemoral means.

RESPONSE
Nemoral is the scientific way to refer to the understorey. However, following the reviewer's suggestion, we have changed this term in the text with the more common term understorey.

Page 3, Line 24: How do the authors calculate the fetch size?

RESPONSE
We intended for fetch the maximum upwind distance from the tower, where surface was homogeneous in composition and micrometeorological behavior. We calculated it as the distance between the tower and the edge of the forest in each direction.
Following the reviewer's remark, in order to clarify, we have decided to substitute the term "fetch" with "distance from the edge of the forest".

Page 3, Line 30: Will the authors please use "instrument models" or "instrument types" here instead of "model"?

RESPONSE
OK, thanks. Corrected in the text

Page 3, Line 31: 10 m away from the tower in the vertical? Or in the horizontal?

RESPONSE
In horizontal direction. A sentence has been added in the text to clarify the direction.

Page 4, Lines 4-10: Are COFA, ROFI, FROM, and GFAS acronyms? If so, it would be helpful to spell out what they stand for.

RESPONSE
COFA, ROFI and FROM are acronyms; we have now specified their meaning in the text.
GFAS instead is the name of a spin-off German company created to develop the instrument which was called with the same name of the company. However, this company is no longer running.

Page 4, Line 19: Why how the data are stored (i.e., in hourly files) relevant?

RESPONSE
Ok, we agree that how the data were collected and stored is not particularly relevant but it was mentioned in order to provide a complete information.

Page 4, Line 26-27: I'm not sure that I understand. Why is the chamber only measured for six minutes? What do the authors use the measurement of the reference chamber for?

RESPONSE
The reference chamber acted as a blank, and it was necessary for the correct flux calculation in the measuring chambers. Please note that after each measuring chamber was sampled, a measurement in the reference chamber was also performed.
For each measuring chamber we had 6 minutes of sampling and 6 minutes of measurement in the reference chamber. Using 5 measuring chambers, the overall cycle lasted exactly 60 minutes (6min x 5 + 6min x 5).
We have inserted a reference to the Bütterbach-Bahl paper where this system is well described and to which the reader could refer for further details.

Page 5, Line 8: I don't think model should be abbreviated, here and elsewhere

RESPONSE
Ok thanks, corrected in the text

Page 5, Line 22-24: The authors should check the consistency of their abbreviations. For example, Campbell Scientific is abbreviated in one sentence but in the next sentence it is not.

RESPONSE
Ok thanks, corrected in the text

Page 6, Line 7-9: What was the result of the inter-comparison between the two COFAs?

RESPONSE
The two instruments were two clones, so we did not expect any particular difference between their measurements. In any case, we decided to run the two COFAs in parallel for 5 days (the typical duration of a coumarin-47 target) in the month following the end of the field campaign.
The figure below shows the average diel course of the deposition velocity measured with the two instruments in the inter-comparison period. A good agreement between the two instruments can be appreciated.

[Figure]

Page 6, Line 10: Are the authors examining the relative standard deviation for a given half-hour?

RESPONSE
Yes, we have considered the standard deviation of each half-hour of the deposition velocity of each of the three instruments.

Page 6, Line 11: Why is systematic in parenthesis?

RESPONSE
Ok, thank you. This was an oversight. Parenthesis have been removed.

Page 6, Line 22: I don't think including this sentence is necessary.

RESPONSE
OK, thank you. The sentence has been removed.

Page 7, Line 9: "were" should be "was"

RESPONSE
Ok, thank you. Corrected.

Page 7, Line 10: What does "semi-hourly" mean?

RESPONSE
It means half an hour: the text was corrected accordingly.

Page 7, Line 13: "k" should be "K" in kinematic

RESPONSE
Ok, thank you. Corrected.

Page 7, Line 26: Is V0 the initial voltage?

RESPONSE
No, $V_o$ is an estimation of the voltage at zero ozone concentration. The text has been corrected to explain this concept.

Page 7, Line 32: The "C" of "C_O3" is missing

RESPONSE
Ok, thanks. Corrected in the text.

Page 7, Line 33: "let" should be "allow"

RESPONSE
Ok, thanks. Corrected in the text

Page 8, Line 17: What is "it" here?

RESPONSE
"It" was meant to indicate ozone. The text has been slightly corrected.

Page 8, Line 7-19: What is the timescale that the authors correct for storage over?

RESPONSE
The timescale used to correct for storage over was every half an hour. We have added this information in the text.

Page 8, Line 24: There is a "the" missing from between "cooled" and "air"

RESPONSE
Ok, thank you, corrected in the text.

Page 8, Line 28: Is this local time? If so, it would be more straightforward to refer to it as local time.

RESPONSE
Yes, it is local time. The text has been modified accordingly.

Page 8, Line 29: Do the authors mean the respective lower and higher levels by "other ones"?

RESPONSE
Yes, that's the correct meaning. The text was slightly modified in order to clarify.

Page 8, Line 29-30: I think the authors mean that only two significant rainfalls occurred during the campaign and they happened at the end of the campaign

RESPONSE
Yes. The text has been modified accordingly to the reviewer's suggestion.

Page 11, Line 10: Why is this remarkable?

RESPONSE
The good agreement between ozone fluxes and LE fluxes (water fluxes) indicates an important role of the stomatal activity in the ozone removal process, and for this reason it should be remarked, even though this topic was not furtherly discussed in the manuscript because it is beyond the aim of this work. The water flux in closed canopies is used as an indicator of the stomatal activity. High LE fluxes = high stomatal aperture.
Anyway, we have removed this adjective and we have slightly modified this sentence in order to be more clear.

Page 11, Line 11: Why would ozone fluxes and LE fluxes be correlated?

RESPONSE

The latent heat flux (water flux) can be considered as an indicator of the stomatal activity in closed canopy. The higher water fluxes the higher stomatal opening. Consequently, higher stomatal opening lead to higher ozone uptake by stomata. Is thus not surprising that LE and ozone fluxes are correlated above a physiologically active vegetation.
Please refer also to the response above.

Page 12, Line 1: There is a rogue "g);" in this line that needs to be deleted

RESPONSE
Ok, thanks. Deleted from the text.

Page 12, Lines 7-8: But there weren't necessarily decreases in emission, there were just decreases in the observed flux (perhaps one could call this the effective emission?)

RESPONSE
We did not completely get the point raised by the reviewer, sorry.
Speaking of soil NO we have used the terms "emissions" and "fluxes" as synonymous because at the soil level NO is always emitted in our ecosystem.
In any case, we have followed the reviewer suggestion by substituting the term "emissions" with "observed fluxes" in the text when appropriated.

Page 12, Lines 9-10: Instead of "followed nearly symmetrical" I would say they were close to inversely proportional

RESPONSE
Yes, the reviewer is right, the term symmetrical should be intended as reflection symmetry. The text has been corrected as suggested, thanks.

Page 12, Lines 11-12: Do the authors mean that the simultaneous minimums?

RESPONSE
No, we mean that the NO fluxes and NO2 fluxes had a simultaneous minimum and maximum, respectively, in the afternoon. The sentence has been modified in order to clarify this.

Page 12, Lines 14-16: The authors should refer to the figures that they are concluding this from. Will the authors please be more explicit about what they mean by "in relation to" here? Why is NO deposition most apparent at this time?

RESPONSE

We have now inserted references to the Figures to support our explanation.
The NO deposition peak above the canopy in the morning can be deduced in Figure 10e (now Figure 9e), black line. The NO deposition peak in the morning can also be deduced from Figure 10b (now Figure 9b) where a small NO gradient can be appreciated and a minimum of the NO concentrations at canopy level can be observed.
This deposition peak is likely related to NO consumption at the top-canopy level early in the morning.
"Why is it most apparent at this time?". We do not exactly know why. However, one possible explanation could be linked to the stomatal uptake of NO2 (which is more soluble in the sub-stomatal cavities water than NO, Teklemariam et al., 2006) by the upper canopy leaves, and this occurs in the early morning when stomata re-open. This NO2 sink would lead to a shift of chemical equilibrium towards an enhancement of the reaction between O3 and NO, thus resulting in the observed O3 deposition peak.
The term "In relation to" was intended as "correlated to". We have modified this term and all the sentence in order to clarify what we mean.

- Teklemariam, T. A. and Sparks, J. P.: Leaf fluxes of NO and NO2 in four herbaceous plant species: The role of ascorbic acid, Atmos. Environ., 40, 2235-2244, 2006.
- Stella, P., Kortner, M., Ammann, C., Foken, T., Meixner, F. X., and Trebs, I.: Measurements of nitrogen oxides and ozone fluxes by eddy covariance at a meadow: evidence for an internal leaf resistance to NO2, Biogeosciences, 10, 5997-6017, 2013.

Page 12, Lines 22-25: Are the authors talking about the decreasing instrument footprint size at lower levels of the canopy? I think their logic needs to be spelled out a little more.

RESPONSE
Yes, the reviewer is right.
A possible explanation of these differences could lie in the different footprints of the eddy covariance measurements. The footprints of the measurements at 41 m, 32 m and 24 m were all falling inside the surface of the upper forest crown, even though 24 m level was just at top-canopy edge. The size of the footprint areas obviously decreased at decreasing measuring heights. However, in absence of any source or sink of the considered scalar, the horizontal homogeneity of the studied ecosystem ensures the validity of the constant flux hypothesis and thus the measurements referred to different footprints should be the same, i.e. fluxes with larger footprints (measurements at 41 m and 32 m) should be comparable to those with smaller footprints (measurements at 24 m).
The above sentences have been added in the discussion section to improve the explanation on the possible role of the horizontal homogeneity and different footprints.

Page 13, Line 12: The authors should be careful as to whether they refer to time as HH.MM or HH:MM, here and elsewhere

RESPONSE
Ok, thanks, corrected in the text. We have chosen the HH:MM notation.

Page 13, Lines 4-21: I find the discussion of the sensible heat fluxes interesting, but tangential. It would be helpful to the reader if the authors did not digress.

RESPONSE
Ok, following the reviewer's suggestion we have decided to remove this part of the discussion. As a consequence, also the Figure 12 was removed because it was meant to support this removed part.

Page 13, Lines 27-34 & Page 14, Lines 1-7: This text should be one paragraph, not three. Also, I would avoid using the word "fact"

RESPONSE
Ok, thank you, the text has been corrected accordingly to the reviewer's suggestions.

Page 14: Lines 1-3: I am not sure I am following this description well. It would be helpful if the authors more clearly spelled out their calculation. Perhaps pointing the readers to compare the grey dashed line and the red line would be helpful for readers.

RESPONSE
Thanks for this suggestion. This analysis is based on the hypothesis that the ozone fluxes measured at 24 m were greater because in the upper canopy (around 24 m) a chemical reaction between O3 and NO took place.
So, supposing a stoichiometric reaction between NO and O3 at the top of the canopy, by subtracting to the 24m O3 fluxes an amount of ozone equal to the NO converging at the top of canopy every half-hour, the part of the O3 flux not caused by this chemical sink is obtained. This is shown in the Figure 13 (now Figure 10) where the measured O3 flux at 24 m is represented as a green line and the resulting part of the O3 flux at 24 m not due to the NO sink is reported as a dark grey dashed line, being the NO fluxes converging from above and below canopy represented by the black and purple lines respectively.
The good agreement between the ozone fluxes at 32 m (Figure 13, red line) and the part of the O3 flux at 24 m not due to the NO sink (Figure 13, dark grey dashed line) suggests that the enhancement of the O3 fluxes observed at 24 m was related to the interactions of O3 with NO at the top canopy level.
We have modified the text following the reviewer's suggestions and indicated the correspondence between the subtracted quantities and the lines in the graph of Figure 13 (now Figure 10)

Page 14, Line 4: instead of "to this ozone sink", will the authors say something like "to

the observed ozone flux"?

RESPONSE
Yes, the referee is right, we have modified the text accordingly to the suggestion

Page 14, Lines 4-5: By "the contribution of the NO deposited from the atmosphere above", are the authors talking about the contribution of NO transported into the forest to the observed ozone fluxes?

RESPONSE
Yes, we meant the deposition of NO transported into the forest from the atmosphere above, as measured at the 32 m level. The text has been changed accordingly.

Page 14, Line 6: What is the "top canopy enhancement of the ozone sink"?

RESPONSE
In this case we meant the fluxes at 24 m which were greater than the fluxes at the higher levels. Thanks for this remark, the text has been corrected to clarify the meaning of this sentence.

Page 14, Line 8: The "of that" is unnecessary

RESPONSE
Ok, thanks. Corrected in the text

Page 14, Lines 8-25: This should be one paragraph, not three

RESPONSE
Ok, thanks. Corrected in the text.

Page 14, Lines 33-34: Please clarify the last calculation - "the residual of the difference of ozone deposition at 5 m and the NO emitted from soil".

RESPONSE
Thanks for the observation, we agree with the reviewer that this sentence was not completely clear. We meant that the ozone deposition to the forest floor was calculated as the stoichiometric difference between the ozone flux at 5 m and the NO flux emitted by soil, namely the amount of ozone which is not removed by chemical reaction with NO.
The text has been modified to clarify the calculation.

Also, do the authors perform this calculation on half-hourly data or on the campaign averages for a given hour?

RESPONSE
This calculation has been performed on the campaign averages for every given hour.

Page 15, Line 3: Specify that it's ecosystem removal relative to air chemistry removal

RESPONSE
Sorry, we are not sure having completely understood this point. Maybe it was not so clear we were addressing the ozone removal by the forest crown.
Roughly speaking, a forest ecosystem = canopy + soil + trunk space air.
In this sentence we were addressing to the canopy component of the ecosystem (excluding the soil and the trunk space air). For this sake we used the term "forest canopy" and not the term "ecosystem" in the first part of the sentence. Instead, in the second part of the phrase we mentioned the term ecosystem referring to all the components.
Following these definitions, the percentage of ozone removal by the forest canopy (upper and lower canopy layers) was nearly 80% of the total amount of ozone deposition measured at 32m. The remaining part (20%) was removed by understorey and soil surface (2%) and air chemistry (18%, reactions with NO).
We have slightly modified line 3 and specified the above percentages in the following text.

Page 15, Line 5: Isn't the depression in the ozone deposition to the dominated crown?

RESPONSE
The reviewer is right, and a midday depression in the ozone deposition can be appreciated also in the dominated crown (lower canopy layer).
However, since this observation is not necessary to the overall discussion we have decided to remove it.

Page 15, Line 14: Specify NO depletion of ozone

RESPONSE
OK, thanks. Corrected.

Page 15, Line 17: What do the authors mean by "stirred"?

RESPONSE
We meant that inside the chambers the air was perfectly mixed during the measurements.

Page 15, Line 19: Why is only a small amount of uptake of NO possible? What are the references for this?

RESPONSE
It is likely that NO2 is deposited but not NO, both due to the fact that the reaction time of NO with O3 is fast and that NO2 has a much higher solubility than NO. Teklemariam et al. (2006) reported that the concentration of NO in the leaf is always larger than 99% of that outside which means no deposition. They report that NO2 is more soluble in water and exhibits a faster hydrolysis rate compared to NO (Henry's law coefficient of NO2 is an order of magnitude higher than that of NO). In Stella et al., for instance, NO2 deposition was detected but no NO deposition was reported.

- Teklemariam, T. A. and Sparks, J. P.: Leaf fluxes of NO and NO2 in four herbaceous plant species: The role of ascorbic acid, Atmos. Environ., 40, 2235-2244, 2006.
- Stella, P., Kortner, M., Ammann, C., Foken, T., Meixner, F. X., and Trebs, I.: Measurements of nitrogen oxides and ozone fluxes by eddy covariance at a meadow: evidence for an internal leaf resistance to NO2, Biogeosciences, 10, 5997-6017, 2013.

The sentence has been slightly modified and these two references have been added.

Page 15, Line 20: What do the authors' findings mean for the canopy reduction factor for NOx?

RESPONSE
The canopy reduction factor (CRF) for NOx is the fraction of NO emitted by the soil that does not exit the plant canopy but is converted to NO2 and then taken up as NO2. Since we argue that all soil NO is transformed to NO2, we would need to estimate the NO2 deposition to the canopy and the NO2 flux above the canopy to correctly estimate that factor. In the absence of NO2 flux measurements, this can only be done by interpretation via a numerical model, which will be the focus of a follow-up paper.

Page 15, Line 27: Which studies are these values comparable to?

RESPONSE
The average maximum of the ozone fluxes was between 10 and 15 nmol m-2 s-1 like, for instance, in Gerosa et.al (2005, 2009a), in Fares et al. (2010) or in Finco et al. (2017). These references have been added to the text, even if usually we do not insert references in the Conclusions.

- Gerosa, G., Vitale, M., Finco, A., Manes, F., Ballarin Denti, A., and Cieslik, S.,: Ozone uptake by an evergreen Mediterranean forest (Quercus ilex) in Italy. Part I: micrometeorological flux measurements and flux partitioning, Atmos. Environ. 39, 3255-3266, 2005.
- Gerosa, G., Finco, A.,Mereu, S., Vitale, M., Manes, F., BallarinDenti, A.: Comparison of seasonal variations of ozone exposure and fluxes in a Mediterranean Holm oak forest between the exceptionally dry 2003 and the following year, Environ. Pollut. 157, 1737–1744, 2009a.
- Fares, S., F. Savi, A., Muller, J.B.A., Matteucci, G., Paoletti, E.: Simultaneous measurements of above and below canopy ozone fluxes help partitioning ozone deposition between its various sinks in a Mediterranean Oak Forest, Agr. Forest Meteorol. 198–199, 181–191, 2014.
- Finco, A., Marzuoli, R., Chiesa, M., & Gerosa, G. (2017). Ozone risk assessment for an Alpine larch forest in two vegetative seasons with different approaches: comparison of POD1 and AOT40. Environmental Science and Pollution Research, 24(34), 26238-26248.

General comments on figures and tables.
Table 1: What does the text in the parentheses mean?

RESPONSE
The text in the parentheses reports the manufacturer of the instrument and the country where it was made.
Pease note that in the revised version of the manuscript Table 1 was moved in the Supplementary material (Table S1).

What do these instruments measure?

RESPONSE
Apart from the first two columns (anemometers and ozone analysers, for which it should be clear what they measure), in the brackets after each instrument/probe we have added a synthetic information on which parameter they measure and an explanation of it was given in the caption.

What height is the nearby mast?

RESPONSE
The height of the nearby mast was 5 m as specified in the paragraph 2.3, however, we have added this information also in the caption which has been completely rewritten.

Figure 5: What are the distributions over?
All the days in the measurement campaign?

RESPONSE

Every average diel course is referred to the second part of the campaign, when, after the intercomparison period, the fast ozone analyzers were deployed at 32 m, 24 m and 16 m.
A sentence was added in the paragraph 2.7 to clarify that the averages are referred to the "profile" period.
In each diel course each point represents the average of all the measurements made in different days at the same half-hour. For instance, the point at 13:30 represents the average of all the measurements made in different days between 13:00 and 13:30.

What is z? What is L?

RESPONSE
z is the measuring height and L is the Obukhov length, this information has been added in the caption

What are the values of z/L used for each class?

RESPONSE
We used the following classification taken from Gerosa et al. 2017, which is an adaptation of the Table from Foken et al. (2008):

| Stability classes | L value | z/L for z=41 m |
|---|---|---|
| Very stable | $0 < L <= 10$ | $0 < L <= 4.1$ |
| Stable | $10 < L <= 100'000$ | $4.1 < L <= 0.00041$ |
| Neutral | $Abs(L) > 100'000$ | $Abs(L) > 0.00041$ |
| Unstable | $-100'000 <= L < -100$ | $-0.00041 <= L < -0.41$ |
| Very unstable | $-100 <= L < 0$ | $-0.41 <= L < 0$ |

This information has been added in the caption of Figure 4.

The height label is missing for plot e).

RESPONSE
Ok, thanks, the information has been added.

Figure 10d: Please indicate the green line is rainfall.

RESPONSE
Ok, added in the caption, thanks.

Also, is rainfall in the context of NO fluxes discussed?

RESPONSE
Yes, it is. The role of rainfalls and rain events in enhancing the NO soil emission was discussed in the context of the NO fluxes.

Are all figures averages over the entire campaign?
If so, please specify in the figure captions.

RESPONSE
Not exactly, all the average diel courses are referred to the second part of the campaign when the fast ozone analyzers were deployed along the tower at different heights. A sentence was added in section 2.7 to specify this information without repeating it in every caption.

**Anonymous Referee #2**

Interactive comment on "Characterisation of ozone deposition to a mixed oak-hornbeam forest.
Flux measurements at 5 levels above and inside the canopy and their interactions with nitric oxide"
by Angelo Finco et al.
Anonymous Referee #2

Close examinations of the pathways controlling ozone deposition in the forest setting
are important for understanding the oxidation chemistry in the forest, secondary organic
aerosol formation, the boundary layer ozone budget, and, as mentioned in this manuscript, the impact of
ozone on plant health. In this manuscript, the authors presented the ozone and NOx concentration gradient
and flux data, and the associated meteorological parameters from the ECLAIRE campaign in 2012, as well
as the initial data analyses, which would lead to a subsequent model analysis, as implied in the Introduction,
that may generate model predictions of intra-canopy dynamics involving ozone reactions with NOx and
VOC. As a result of the month-long summertime observations of ozone and NOx at multiple height within
and above a forest canopy at the most polluted area in Europe, this dataset provides a valuable case study
of ozone dynamics and related canopy scale processes, and biosphere-atmosphere exchange. However,
as specified below, major revisions are recommended.

The authors performed a fairly detailed treatment of the meteorological data to obtain the ozone fluxes at
the measurement heights. Results of the fluxes at multiple elevations throughout the canopy provide
information on the sources and sinks, thus the processes that affect the trace gas species in the forest.
Given the data being from a month-long campaign under different meteorological conditions, the analysis
could be strengthened and the conclusions may be better supported and possibly modified with the
following additional considerations.

RESPONSE
We thank the reviewer for the careful revision and his/her suggestions to improve this manuscript.

1) Above canopy influences from air transported to the surface layer above the canopy.
According to the data, about 50% of the wind is from either east or west with the rest from other directions.
Are there any differences in the quantities measured that coincide with the wind direction differences? Are
the possible influence of the different amount of the pollutants (ozone and NOx) transported to the site
considered?

RESPONSE
Thanks for this suggestion. We have explored this possibility, but we did not find significant results that
might explain, for instance, the greater ozone fluxes at 24 m. The behavior of the NOx and O3
concentrations was similar regardless of the wind provenance, so we have focused our attention on the in-
canopy processes. For example, here below we report the average diel course of ozone and NOx
concentrations above and below canopy when the wind was blowing from east (right graphs) or west (left
graphs). Obviously there are some differences, but they are very weak and not relevant. For instance, O3
maximum was slightly higher when the wind was blowing from east, while NO was 1 ppb higher in the
morning peak when the wind was blowing from west, and also the NO$_2$ peak was lower in the morning (-4

ppb). However, since the "vertical" behavior of the NOx concentrations was similar independently from the wind direction (NO concentrations had a minimum at the canopy level height while $NO_2$ concentrations were almost constant at all heights) we concluded that the peculiar behavior showed by the fluxes at 24 m was more linked to processes happening in the forest rather than to advection of pollutants from different directions.

[Figure]

*note: right x-axis is referred to the NOx measurements at soil level (15 cm)

2) The fluxes under the stable/unstable conditions within the canopy. If I understand it correctly, in Figure 4, y-axis is the percent of the measurement time. If so, there were times the entire canopy was either stable or unstable throughout a 24-hour period. It would be informative to separately analyze the data under these two regimes, especially when considering the within the canopy stability in the context of the enhanced ozone deposition flux at 24 m.

RESPONSE
We are not sure we have understood the reviewer's point.
Figure 4 report the percentage of the times that at a given half-hour there was a stable/unstable atmosphere at a certain level during the measuring campaign.
That being said, from the morning to the late afternoon at 41 m, 32 m and 24 m there were nearly always unstable/very unstable conditions. At 41 m only 3.3% of the measurements were characterized by stable conditions between 8:00 and 18:00 (Figure 4a) and this percentage was only 4.1% at 32 m (Figure 4b) and 19.4% at 24 m in the same hours.
Stable conditions in the central hours of the day were more frequently observed only for the lower measuring points (16 m and 5 m), so in our opinion there were too few data to significantly compare for contrasting conditions of the atmospheric stability on the entire canopy and above, and to verify whether this influenced the ozone deposition flux at 24 m, sorry.

3) Dry and wet conditions. Apparently after the rainfall on July 6, the NO/NO2 fluxes increased dramatically. How did these changes affect the ozone deposition flux?

RESPONSE
Thanks for the comment and the suggestion. The peak of the ozone fluxes at 24 m increased by 70% after the rainfall in the morning hours, as reported in the Figure below, where the average diel course of the ozone fluxes of 3 days before rainfalls (from 4th to 6th July) and 3 days after (from 8th to 10th) are shown. This is consistent with the hypothesis that the flux enhancement at 24 m is linked to the NO flux from the forest floor. Instead, the ozone fluxes at the other levels remained more or less unchanged, with the only exception of the flux at 32 m which was never greater than 5 nmol $m^{-2}$ $s^{-1}$ as absolute value from 9.00 to 12.00.
However, since only one rainfall event occurred during the field campaign, we have preferred not to include this analysis in the manuscript because of the questionable representativeness of our data with respect to this condition.

[Figure]

[Figure]

In addition, it is known surface wetness affect ozone deposition as well. It would be instructive if the dry/wet conditions can be contrasted in the data analysis.

RESPONSE
We have also considered the possible influence of surface wetness on the O3 flux deposition, but we have excluded a significant role of water/dew because the canopy was almost always dry as witnessed by the dew temperature which remained well below the air temperature, as it can be observed in the figure reported below.
Moreover, the relative humidity at canopy level (24 m) was almost never above 90% (less than 1% of the data), thus excluding the possibility of dew formation on leaves and branches.
Also in this case we have preferred not to include this analysis in the manuscript because of the questionable representativeness of our data with respect to this condition. Moreover, we would like to avoid to enlarge excessively the paper, which has been already considered too long.

[Figure]

The data analysis results, as stated in the manuscript now, would be more convincing if the above-mentioned aspects were considered. One of the main conclusions in the manuscript is that the enhanced ozone flux at the canopy top is due to the combined NO fluxes from above the canopy and the soil emission, thus an enhanced chemical sink of ozone. However, there are other factors such as stomatal uptake and photochemical reactions involving NOx, O3 and BVOCs, both processes obviously associated with sunrise, that affect the ozone flux. The net result could well be an increased flux. Are there data available from this campaign that would help the authors address these possibilities?

RESPONSE
We agree with the reviewer.
However, BVOC dynamics during this field campaign was already described in Schallart et al. (2016) and Acton et al. (2016). Preliminary results from Nemitz et al. (2013) showed a nearly negligible role of the BVOCs on the O3 deposition fluxes, since less than 3% of the deposited ozone was destroyed by reaction with BVOCs. As a consequence, we can exclude a significant role of BVOC emissions and have not included this topic in this manuscript.
Moreover, the role of the BVOCs and their interactions with ozone will be dealt in detail in another future paper more focused on the chemistry from a modelistic point of view.
Stomatal ozone flux partition was not presented in the paper because evapotranspiration fluxes were available only at the topmost level of the tower (41 m). Since we do not have evapotranspiration

measurements at 32 m, 24 m and 16 m we could not evaluate the stomatal contribution to the ozone fluxes at the different canopy layers with the usual stomatal flux partition procedure. Instead, we preferred to focus our manuscript on the attribution of the total ozone deposition to the different canopy layers. In our humble opinion, including a stomatal flux partition referred to the whole canopy along with the attribution of the total ozone fluxes to the different canopy layers (upper, lower canopy layers, understorey, forest floor), could have been confusing for the reader.

Moreover, the inclusion of a new section on stomatal fluxes would have made the manuscript longer, particularly because a long description of the partition methodology and calculation should have been added to the Material and methods.

However, more than one new papers dealing with the ozone stomatal fluxes at the same site are in preparation and will cover longer measuring periods, namely annual and pluriannual duration.

It would help the readers to better understand and assess the data if the authors could present the time series plots, including error bars when appropriate, of the measurement results.

RESPONSE
Sorry, we are not sure we have understood the reviewer's point.
We have already shown the time series plots of the most important variables like the ozone concentrations and fluxes, the NO and $NO_2$ fluxes, LE fluxes, air temperature at different levels and rain. However, the reviewer #1 requested to move many figures to the supplementary material because of the high number of Figures presented in the manuscript. Following his suggestion, we have preferred to move the above-mentioned Figures based on raw time-series to the supplementary material.
Regarding the error bars, maybe we misunderstood, but they do not apply to the raw time-series because every point there represents one measurement only.
Instead, for the diel courses where each point represents the average of many measurements, please refer to the following answer.

It is also important to show the standard deviation (if the mean values are used) or the interquartile range (if median values are used) in the average diurnal course plots including Figure 6.

RESPONSE

We have already posted two comments in the discussion about this referee's remark. In our humble opinion we think that in many figures of our manuscript, because of the presence of five lines (or more) in the figures, the use of error bars could be misleading for the reader because error bars are overlapping and could hide the diurnal course of the plotted parameters.

For example, please refer to the following graph reporting the diel course of [O3] at six levels. The error bars hide the course of the ozone concentrations at the different heights when the lines are close. For example, it is nearly impossible to understand the course of the ozone concentrations at 41 m.

[Figure]

However, we agree with the reviewer that the explication of the uncertainties of the means is of primary importance. So, we would suggest to report the range of the variation of the standard error for each curve (from the minimum to the maximum values observed in the 24h) in the text of the figure captions, without losing statistical robustness and the possibility to clearly appreciate the diel courses.

The following Figure reports and example of how we cope with this issue throughout the manuscript.

[Figure]

**Figure 2 - Average diel courses of ozone concentrations at the six levels (41 m, 32 m, 24 m, 16 m, 5 m and 0.15 m). The maximum and the minimum standard error of the half-hourly means were respectively 3.0 and 1.7 ppb for 41 m, 3.0 and 1.8 ppb for 32 m, 3.4 and 1.9 ppb for 24 m, 3.4 and 1.8 ppb for 16 m, 2.9 and 1.7 ppb for 5 m, and 3.4 and 1.0 ppb for 0.15 m**

In Table 1 where it is not indicated or obvious, please list the measurements next to the instruments listed, for example, HMP45 (temperature, humidity).

RESPONSE
Ok, thank you, Table 1 has been rearranged and now include all the requested data.
Pease note that in the revised version of the manuscript Table 1 is now available in the Supplementary material (Table S1).

I may have missed the point in Figure 8 but cannot readily see from which height are the plotted ozone mixing ratio results.

RESPONSE
Ok, the reviewer is right. We have added the measuring height of the ozone concentration data in the caption.
Pease note that in the revised version of the manuscript Figure 8a is now Figure 7 and Figure 8b has been moved to the supplementary material (Figure S3).

Also if possible, please unify the tick location and tick labels in plots 8a and 8b.

RESPONSE
Ok, thanks for the suggestion. Figure 8b has been moved to the supplementary material (Figure S3) and the time resolution of the x-axis was set like in Figure 8a.

I am not sure why the analysis of the enhanced O3 flux at 24 m is only for the morning (9:00-12:00). From the data, the enhancement, although decreasing after mid-day, lasts through 15:00. If, because of the scatter of the data, the fluxes were basically the same from 41 to 24 meters in the early afternoon, it needs to be shown and stated more clearly.

RESPONSE
The reviewer is right and we would like to thank him for this observation because he allowed us to realize that we made a mistake in assembling the final version of the manuscript.
Unfortunately, a previous version of Figure 9 (now Figure 8) remained in the submitted document. This previous version reported the ozone fluxes calculated by applying an average Frequency Loss Correction (FLC) factor (one for each level) coming from a preliminary elaboration made with the ogive methodology (OG).
Instead, in the correct version of Figure 9, that we report below (9a, left graph), the fluxes were corrected for the high frequency flux losses by applying, half-hour by half-hour (for each level), the Experimental Transfer Function methodology (ETF) described by Aubinet et al. (2000, 2001, 2012).

**9a – Correct version of Figure 9**

**9b – Previous (wrong) version of Figure 9**

[Figure]

[Figure]

We abandoned the FLC initially calculated with the OG methodology because we realized that the ogive of the 24 m data (and only the 24 m ogive, not the others), beside showing the highest correction factor presented a relatively strange shape: the ozone ogive was not a monotonic increasing function but showed a relative maximum around 0.1 Hz and a following inflection between 0.1 and 1 Hz, as it can be observed in the following Figure. This feature could have resulted in an incorrect evaluation of the FLC factor.

[Figure]

**Figure - Normalized ogives of ozone and sensible heat fluxes for the measurements at 24 m.**

The ETF methodology on the contrary, is less affected by subjective interpretations as it is more easily automatable and applicable to each half-hour sample. For this reason, it was preferred to the OG methodology.

The application of the ETF correction resulted in a lesser pronounced overestimation of the 24 m ozone fluxes than with the OG methodology (Figure 9a and 9b), but the ozone fluxes of the other levels resulted relatively unaffected by this change of methodology, as reported in the following Table (where the overall average of the FLC factors after the application of the two methodologies are indicated).

| level | **FLC applied** | |
|---|---|---|
| | Figure 9a (ETF corrected)* | Figure 9b (OG corrected)** |
| **41 m** | 1.08239 | 1.01012 |
| **32 m** | 1.00912 | 0.98030 |
| **24 m** | 1.06197 | 1.18723 |
| **16 m** | 1.02083 | 1.06385 |
| **5 m** | 1.03198 | 1.03561 |

*The reported values for each level are the average of the FLC values calculated for each single half-hourly sample (i.e. each sample had its own correction factor).
** The reported values for each level are the single correction factors applied to all the half-hourly samples (i.e. the same factor was applied to all the half-hourly samples).

In order to demonstrate that it was just an oversight and a careless mistake we invite the reviewer to draw his attention on Figure 13 of the paper (now Figure 10) which was already based on the ozone fluxes ETF corrected.

By comparing the 24 m fluxes in Figure 13 with the 24 m fluxes of the Figure 9a (correct version) the reviewer can realize that the O3 fluxes at 24 m are the same (green lines), regardless of the different

averaging time (the time scale of the Figure 9a is half an hour - the time scale of eddy covariance fluxes - while the time scale of Figure 13 is one-hour - the time scale of the NO soil fluxes - and thus the eddy data were averaged on an hourly base).

We add here the Figure 13 of the manuscript for comparative purposes. As the reviewer can see, the peak values of the green lines in these two figures are the same (-13 nmol m-2 s-1) because Figure 13 was indeed made using the data of the 24m fluxes calculated with the ETF methodology.

[Figure]

*Figure 13 Average diel course of ozone fluxes at 32 m (red line), ozone fluxes at 24 m (green line), NO fluxes at 32 m (black line) soil NO fluxes (purple line) and modified ozone fluxes at 24 m (dashed grey line). This latter takes into account the role of the NO sink.*

Now, considering the correct version of Figure 9 (Figure 9a), the reviewer can understand what we meant in the following sentence of the discussion:

"…Fluxes of ozone were similar at these two above-canopy heights, within the uncertainty of the measurement, but unlike for the fluxes of heat and momentum, measurements at 24 m showed a significantly larger ozone deposition than the two upper levels (32 m and 41 m) at certain times of the day. In the morning (9:00 to 12:00) 24 m ozone fluxes were on average nearly 3 nmol m-2 s-1 larger than above the canopy (Figure 9), while they were nearly equal on average from 13:00 to 18:00 (fluxes at 24 m were only 0.5 nmol m-2 s-1 larger)…"

Moreover, we would like to underline that the core of the discussion, i.e. the influence of the NO fluxes on 24 m fluxes, and the subsequent considerations were based on the analysis of the ETF correct data. We thoroughly apologize for that.

The authors used Figure 12 to explain the effect of the thermal bubbles and why the greater sensible heat flux at 32 m than at 41 m. However, the data shown are from 13:00 on July 5th. It is not clear whether this is a special case or an example of a typical situation.

RESPONSE
Figure 12 was just an example of a typical situation which was observed quite often in the temperature raw data.
However, we agreed with the reviewer #1 that, even though interesting, this part of the discussion should be removed in order to avoid excessive digressions from the main topic (ozone and NOx interactions). As a consequence, Figure 12 and the discussion on the sensible heat fluxes at 32 and 41 m have been removed from the manuscript.

The manuscript could use some help with the English language usage and organization.
For example, Page 2, line 4, "Prompted by its phytotoxicity" –> Because of the phytotoxicity of ozone, . .

RESPONSE
Ok, thanks. This sentence has been rephrased.

Page 2, line 6, ". . ., 2013), thanks also to the. . .." may be changed to, for example: . . ., 2013). Eddy covariance measurements were made possible thanks to the . . .

RESPONSE
Ok, thanks. Corrected in the text.

Page 2, line 16 – 18 and line 26-28, field campaign information was repeatedly given.

RESPONSE
Ok, thanks. The second sentence has been removed.
The Introduction has been improved and expanded according to the suggestions of the reviewer #1.

[revised manuscript text omitted]

Figure S1 – (a) Rainfall amounts and temperature evolution at the seven heights. Blue lines are rainfalls. (b) Average diel course of air temperature at the seven heights. (c) Wind rose based on 41 m data, the radial axis unit indicates the percentage of the data in each direction, the blue line diurnal data, the red line nighttime data. (d) Average diel course of wind intensity at the five heights. For figures a), b), d) and e) the numbers in the curves label of the legend represent the measurement height (in meters). For figure b) The maximum and the minimum standard error of the half-hourly means were respectively 0.65 and 0.32 °C for 41 m, 0.66 and 0.33 °C for 32 m, 0.73 and 0.33 for 24 m, 0.71 and 0.31 °C for 16 m, 0.68 and 0.30 °C for 11 m, 1.10 and 0.65 °C for 1.5 m, 1.11 and 0.66 °C for 0.15 m. For figure (d) The maximum and the minimum standard error of the half-hourly means were respectively 0.32 and 0.12 m s$^{-1}$ for 41 m, 0.28 and 0.08 m s$^{-1}$ for 32 m, 0.05 and 0.01 m s$^{-1}$ for 24 m, 0.02 and m s$^{-1}$ 0.01 for 16 m, 1.11 and 0.03 m s$^{-1}$ for 5 m.

[Figure]

**Figure S2 - Mean diel evolution of the ozone storage flux (the storage term of the Eq. 3). The contribution of the air column between adjacent flux measurement levels is indicated with a different color.**

[Figure]

**Figure S3 - Latent heat fluxes measured at the top of the tower (LE 41) and soil water content (SWC) expressed as volumetric ratio between water and soil.**

[Figure]

**Figure S4 - Average normalized cospectra of the vertical component of the wind and ozone at 11:00 (a) and at 15:00 (b).**

---

## Author Response (AR2)

REVIEWER #1

Finco et al. present a new, valuable dataset and have interesting and original findings. Although I do find the paper improved in the first round of review, I still find that the paper lacks cohesion and pretty severely, broader context. The paper is very descriptive, but it is not always explicitly written in the manuscript what the point of the description is and why findings are important. To be published in ACP, I think the manuscript needs to improve in this capacity, so I suggest that the paper still requires major revisions.

RESPONSE
Thank you for these interesting suggestions. As recommended by the reviewer, we have tried to better highlight the importance of our results for a broader research context. We have added in several sections of the manuscript (Abstract, Introduction, Discussion) some sentences on the importance of our measurements for the improvement of multilayer canopy models and models trying to represent the in-canopy dynamics of reactions between $O_3$, BVOCs and $NO_x$. We have also highlighted that these models are important for the correct characterization of the forest-atmosphere gas exchange which in turn may influence air-quality of the peri-urban environment and the forest ecosystem productivity, especially when they involve phytotoxic pollutants such as $O_3$ and $NO_x$. Hopefully, these integrations should help the reader to understand the scientific context of our measurements and their importance. We have also highlighted that this joint measurement campaign provided for the first time O3 flux measurements on a mixed oak-hornbeam mature forest at 5 levels along the vertical profile of the forest ecosystem.

The below comments are minor revisions.

I think something new from the last round of review is that the authors added that one of the goals of the study is to test the constant flux theory (see sentence on end of page 1 to beginning of page 2). Although I think the authors should keep discussion of the constant flux theory in the results/discussion, it seems like it was added last minute as a goal. I would recommend either better integrating this into the study, or cutting it as a goal and keeping the brief discussion of it in the results/discussion.

RESPONSE
As suggested by the reviewer, we have removed the test of the constant flux theory as one of the goal of the manuscript and kept the brief discussion on this issue in the results/discussion section.

There are still too many figures. I would suggest moving Figure 6 to supplemental, especially because this is showing the same data as Figure 5. Figure 7 could be also moved to supplemental. I would also urge the authors to make the figures higher quality. For example, the text on the legends on each plot could be cleaner (have subscripts instead of "_", "m" after the numbers to indicate height). The combination of plots for NOx fluxes and concentrations could also be a lot cleaner.

RESPONSE
Ok, thank you for this suggestion. Figure 6 and 7 have been moved in the supplementary material. Figure 6 is now Figure S2, while Figure 7 is now Figure S4a.
The text of the Figures legend has been fixed: we have deleted the underscores, added subscripts where applicable and "m" after each height value.
NOx plots in Figure 7 (it was Figure 9 in the previous version of the manuscript) have been recombined with the same width. Now the plots combination is cleaner.

Also why are there temperature gradients shown, instead of day-night cycles? It seems like the authors should be consistent with the display of other micrometeorological quantities shown.
RESPONSE
We chose to report the temperature gradients because they easily show the change of the height of the thermal inversion during the average day of the measuring campaign.
The day-night cycles of the temperatures at the different levels were reported in the first version of the manuscript, however during the first revision we were asked to move some of the Figures in the supplementary material and those plots are now in Figure S1b. A more detailed focus on those measurements is also contained in Figure S1a.

I think the error bars on the plots are necessary. There are other options for trying to make the lines on the figures stand out more. For example, the authors could slightly offset the error bars from each other (for example, plotting the data at heights 1,2,3,4,5 at (HH-1):56, (HH-1):58, HH:00, HH:02, HH:04, respectively). Another option is for the authors to reduce the width of the error bars, without reducing the width of the line. If the program they are using does not offer this option, the authors could first plot the line with the error bars using a thin line width, then plot the line without error bars with a thicker line width.
RESPONSE
Ok, thank you for the suggestion. We have added the error bars of the means to the plots when this was applicable. We have inserted the errors bars with a thin line width in order to keep the main lines still readable.

Table S1 is very helpful for readers - I think it should be in the main text.
RESPONSE
Ok, thank you. Table S1 has been moved in the main text as requested, and is now Table 1.

Line 22: Please cut "among them" as this is implied
RESPONSE
Ok, done.

Line 23: Specifying "24 m" is not helpful here because the reader does not know the heights of the other instruments. Please cut "24 m".
RESPONSE
Ok, done.

Lines 27-30: Thanks to the authors for including this in the abstract. I would like the authors to clarify here that some of the ozone "removed by the canopy" is removed by ambient chemistry as opposed to ozone dry deposition on physical surfaces (for example, leaves, bark, and soil). Please also do this in the discussion and conclusion.
RESPONSE
Thanks for this comment. We agree with the reviewer. We have modified the text clarifying this concept in the Introduction, Discussion and Conclusions sections.

Line 31: I think "capability" works better than "capacity" in this context. Technically, ozone dry deposition models don't have ambient chemistry in them (but chemistry models that have ozone deposition models in them do). I would recommend making the part of the sentence that says "with particular regard … NOx" a separate sentence about multilayer canopy models that have representations of both ozone dry deposition and ambient chemistry that are used to tease apart the influence on ambient chemistry vs. ozone dry deposition on observed ozone fluxes.
RESPONSE
We agree with the reviewer. We have added a separate sentence highlighting the importance of this dataset for the multilayer canopy models which include ambient chemistry. We liked the sentence proposed by the referee and reported it in the abstract.

The first couple paragraphs of the introduction should be combined into 1-2 paragraphs
RESPONSE
Ok, thank you, the two paragraphs have been combined.

Line 6: Please cut "functional for such measurements" as this is implied
RESPONSE
Ok, done.

Line 11: "leaves" should be "leaf" or possessive
RESPONSE
Ok, we have corrected the text.

Line 12: "ozone fraction" should be "amount of ozone"
RESPONSE
Ok, we have corrected the text.

Line 13: "starting from" is ambiguous. I would recommend saying something like "For example, soil water availability is positively correlated with .."
RESPONSE
Ok, thank you. We slightly modified the sentence adding "such as, for example,"

Line 15: "the other deposition pathways" should be something like, "ozone deposition pathways other than plant stomata"
RESPONSE
Ok, we have modified the text.

Line 16: "in deep" should be "fully"
RESPONSE
This term has been already modified according to the suggestion of reviewer 2.

Line 18-19: Please clarify that chemical reactions between ozone and NO and BVOCs are ambient reactions, and articulate here that technically these reactions are not really ozone dry deposition, but they influence the observed ozone flux, which is one reason why we care about them. Another reason we care is secondary organic aerosol (SOA) formation (see Goldstein et al. 2004, Wolfe et al. 2011). I think it framing the work in this manner and discussing SOA would help contextualize the importance of this study.

RESPONSE

Ok, we understand the reviewer point and we agree. We have mentioned separately the $O_3$ dry deposition on surfaces and the $O_3$ chemical consumption processes in the sentence before the list of the different processes.

Line 20: Please clarify what about ozone deposition below the forest canopy is unclear — this statement is a bit too vague.

RESPONSE

Thanks for the comment. We have changed the sentence as follows:

"In addition, the $O_3$ deposition dynamics below the forest canopy and their relationship with the above canopy $O_3$ fluxes are still to be fully understood, since only a few studies have directly measured $O_3$ fluxes below the canopy"

Line 20: Please cut "actually"

RESPONSE
Ok, done.

Line 25-27: Will the authors articulate in the text whether their having 5 ozone eddy covariance fluxes in & above the canopy is new? Did Dorsey et al. and Foken et al. have this many fast sensors, or only two? Please clarify. To my knowledge, this is something that has never been done before and should be clearly articulated.

RESPONSE

Thanks for the comment. Dorsey at al. measured at 3 levels (2 above and 1 below the canopy) while Foken et al. (2012) had a 4-level system: 1 above the canopy, 1 at the top-canopy, 2 below the canopy even though no $O_3$ data have been presented in their comprehensive 2012 paper.

Also to our knowledge no other campaign reported 5 levels of $O_3$ fluxes measurements along the vertical profile of a forest.

Lines 27-28: I would argue that canopy profiles of ozone concentrations have been investigated more than fluxes, but there still has not been that much investigation of them (unless they were used to derived fluxes). In part, this may stem from lack of knowledge about in-canopy sources and sinks of ozone (which your study attempts to quantify). Please re-phrase, and consider using this as motivation for your work.

RESPONSE

Thank you for the suggestion. We have re-phrased this paragraph. However, according to the reviewer #2 ozone concentrations have been extensively investigated both above the canopy (for flux calculation with aerodynamic method) and below the canopy (for the characterization of air chemistry dynamics). We have tried to highlight these aspects in the text which could also motivate our study.

Line 14-15: Replace "is represented by" with "is". There is an extra comma after area. This description of a climax ecosystem does not make sense to me. What is a maximal stage of development? In the response to review I see that the authors have include "(mature)" - it would be helpful to describe the forest as "mature" in the text.
RESPONSE.
Ok, thanks for the suggestion. We have removed the description of a climax ecosystem and specify that the site was a mature forest.

Line 32: Please spell out "E" and "NE" before abbreviating for the first time.
RESPONSE
Ok, done.

Line 20: I don't think "East" and "West" should be capitalized here. Please spell out north and south.
RESPONSE
Ok, done.

Line 21-26: Please spell out why evapotranspiration and ozone flux would be correlated (please do this in the text, not only the response to review)
RESPONSE
Ok, thanks. The water flux in closed canopies can be used as an indicator of the stomatal activity of plants. We have added in the text the following sentence explaining why evapotranspiration and $O_3$ flux are correlated:
"The good agreement between $O_3$ fluxes and LE (water) fluxes (Figure S4) indicated the important role of the stomatal activity in the $O_3$ removal process because stomatal opening can increase both transpiration and $O_3$ stomatal uptake"

Line 1: "had" should be "has"; cut "most"
RESPONSE
Ok, done

Line 7: specify "because both heights are above the forest canopy" after "almost equal"
RESPONSE
Ok, done

Line 8: Why do the sources and sinks for heat and momentum matter for ozone fluxes? Please spell this out in the text (not only in the response to review)
RESPONSE
From a theoretical point of view, the sink for momentum and the source for the heat may influence the $O_3$ deposition to the canopy. In fact, a greater sink for momentum results in a greater probability

for $O_3$ molecules to impact on the plant external surfaces, thus enhancing the physical deposition of $O_3$.
In the same way, a warm canopy enhances the thermal vertical air movements (turbulence) and the probability of the collisions. Moreover, warmer surfaces imply a greater thermal decomposition of $O_3$ on them or in the surrounding air.
We took in consideration the possibility to insert some considerations regarding this issue in the discussion as suggested by the reviewer. However, we found that it would have interrupted the logical flow of the arguments of the discussion which was mainly focused on $O_3$ fluxes rather than momentum and heat fluxes.

Line 18: What difference are the authors talking about here? The ozone flux difference at 32 and 41 m vs. 24 m, or that the authors see a gradient for ozone fluxes but not heat or momentum? Please specify in the text
RESPONSE
We were speaking of the differences between the "enhanced" fluxes at 24 m and the above ones. We have slightly changed the sentence to clarify what we meant.

Line 26: Related is spelled incorrectly
RESPONSE
Ok, thank you, we have corrected the text.

Lines 27-32: These paragraphs should be combined with the paragraph above
RESPONSE
Ok, done.

Lines 4-5: It would be helpful if the i.e. statement was in parenthesis after "..of the canopy"
RESPONSE
OK, done.

Line 9: Note missing subscript on ozone
RESPONSE
OK, done.

Line 10: Note use of "ozone" instead of "O3" - the authors should be consistent throughout the text this this. I see several other instances with "ozone"
RESPONSE
Ok, thank you. We have checked throughout the manuscript and replaced the word "ozone" with $O_3$, with the exception of the sentences in which the term "Ozone" was at the beginning of the text.

Line 9-11: I'm wondering if the authors would consider re-writing this sentence in the active voice - as it is, it is difficult to understand.
RESPONSE

Ok, thank you, the sentence has been changed in active voice as suggested by the reviewer.

Line 12: I'm not sure what the authors are getting at here with "being" - please re-phrase
RESPONSE
Ok, thank you. We have slightly modified the sentence to clarify what we meant.

Lines 14-16: There is not good agreement at night though - why?
RESPONSE
We are aware that during the night there was no agreement between the "corrected" $O_3$ fluxes at 24 m and the $O_3$ fluxes measured above the canopy.
The methodology we have proposed to explain the enhancement of the 24 m $O_3$ fluxes implies that the NO emitted at soil level is transported up to the top of the canopy by the in-canopy mixing processes. However, at night there might be an over correction of the O3 flux at 24 m (Figure 8) because, in case of high atmospheric stability, the NO emitted from soil could stratify near the forest floor and react below the canopy.
The reviewer's remark convinced us to highlight this discrepancy in the text of the discussion.

Line 3: Cut "between them"
RESPONSE
Ok, done

Line 5: I would recommend talking about how no measurements of this type of forest were found elsewhere in the literature in a separate sentence of motivation for the authors' study. For the most part the studies mentioned are for other Mediterranean/Italian forests. Can the authors specify this, and if there are any implications to this comparison? There are a lot more studies that quantify ozone fluxes so it is unclear to me if there is any more meaning to the authors' choice of citations here. If not, I would recommend using "e.g." before the citations listed.
RESPONSE
Ok, thank you for this suggestion. We have added a sentence that highlights the lack of measurements on this type of forest, and new references to studies on ozone fluxes performed in other forest types.

Line 15: "modelists" should be "modelers".
RESPONSE
OK, done

I'm not sure what the authors are trying to say with this statement. Will the authors please re-phrase? One thing to consider would be: why would modelers want to use this data? What about this dataset is useful in particular? I think the authors should consider context broader than "correctly reproducing intra-canopy dynamics" as stated in the next sentence. For example, why would modelers want to reproduce intra-canopy dynamics?
RESPONSE
Ok, thank you for this suggestion. We have expanded this sentence and highlighted the importance of this measurements for broader context such as air quality, forest-atmosphere exchange dynamics and ecosystem productivity.

REVIEWER #2

This revised manuscript reads much better than the last one - the objectives, main points and conclusions are presented more clearly. The data set is the results of the first multi-level ozone flux measurements in a deciduous forest. Although more detailed analyses are needed to carefully examine the various drivers of ozone dynamics, it is valuable to present the data set and make it available for the next-step model studies.
Below are a couple of remaining questions:
1) The authors' reply to the question about the possible role of BVOC in ozone deposition is reasonable, but why not include the response in the manuscript? In the Introduction, chemical reactions with biogenic volatile organic compounds is listed as one of the non-stomatal deposition processes. When trying to understand the enhanced ozone deposition flux at the canopy top during morning hours, the possible role of BVOC would always come up. It would be helpful to briefly state the results of prior investigations on it and cite the references.
RESPONSE
Ok, thank you for the suggestion. We have inserted in the discussion a new paragraph describing the possible role of the BVOCs in the O3 fluxes dynamics and some references to previous investigations on this issue.

2) One of the main conclusions of the manuscript is that the soil NO emission and NO from the photolysis of NO2 are responsible for the observed additional flux at 24 m compared to that at 32 m, with the former being the larger factor. Based on the ozone flux and mixing ratio data in Figures 8 and 6, it appears the ozone deposition velocity is highest at 24 m and decreases with height. This implies the reaction of the soil-emitted NO and ozone happens mainly near the canopy top, not lower and closer to the forest floor. Given the reported ozone mixing ratio, the lifetime of NO is on the order of 100 s. This is to say that the turbulent mixing within the canopy needs to be on the time scale that NO is transported to the upper part of the canopy but not ventilated to the atmosphere above. Do the data support this?
RESPONSE.
The reviewer's observation is interesting.
We have checked our data. Although direct fast measurements of NO inside the canopy and along the vertical profile (i.e. between 24 m and 0.15 m) were not performed, we have tried to indirectly estimate the transport time of the NO emitted from soil by assuming that turbulence acts in the same way for all the transported scalars. Thus, we looked at the heat transport below the canopy. Interestingly, we have verified that the cospectra of $w$ and $T$ (which integral gives the heat flux) in the morning (e.g. 11:00) showed a peak between 0.010 and 0.015 Hz ($\tau \approx 70\text{-}100$ s) for both 24 m and 16 m levels. These peaks were absent in the afternoon (e.g. 15:00).
This evidence may confirm that the time scale of turbulent mixing below the canopy in the morning hours could be comparable to the lifetime of NO, but this does not exclude that the NO emitted from soil could also react within the canopy or even in the lower part of it.
Moreover, the transport mechanisms inside the canopy are quite complex and not completely clear, as underlined by Ganzeveld et al. (2015):
"…*the renewal of the canopy air, and hence the mass and heat transport, during certain conditions can be almost fully attributed to coherent turbulence structures typically of the size of the canopy that periodically (in the order of minutes) enter very quickly the canopy layer from above. This latter phenomenon is critical in terms of modelling in-canopy transport of reactive species also since this introduces a non-linear relationship between the concentration and the flux and leads to non-diffusive transport and counter-gradient fluxes. The concentration is representative of calm episodes while the flux is driven by short and intensive exchanges. **This makes it also difficult to define a single transport time scale**.*".

This will be a challenge for the modelers that will use our dataset for their simulation, because it will require the development a multilayer model which takes into account the occurrence of discontinuous or periodical exchange processes like sweeps and coherent structures, as well as the in-canopy air chemistry dynamics.

Detailed comments:

Would it be possible for one or more of the coauthors to go over the English for this manuscript and fix the remaining problems?
RESPONSE
Ok, thank you for the suggestion. We have checked the English of the manuscript and tried to improve it and fix the language problems.

Page 1, line 26: is it "morning coupling between the forest and the atmosphere" or rather, the mixing within the canopy?
RESPONSE
The reviewer is right, we have changed this sentence according to the suggestion. However, the two mechanisms (coupling forest atmosphere and mixing within canopy) are obviously strictly linked.

Page 2, line 3: "2018)," should be 2018). – change the comma to a period at the end of ")".
RESPONSE
Ok, thank you. we have corrected the text.

Page 2, line 4-5: The deposition of O3 on forest ecosystems has been extensively studied over the last 20 years with eddy covariance field campaigns (Padro, 1996; Cieslik, 1998; Lamaud et al., 2002; Mikkelsen et al., 2004; Gerosa et al., 2005, 2009a, 2009b; Launianen et al., 2013) which were made possible thanks also to the development of fast ozone analysers, functional for such measurements.
RESPONSE
Ok, thank you. We have corrected the text.

Page 2, line 7-8: Change "while more recently campaigns had extended the observation periods and…" to "while more recently campaigns have extended observation periods and…"
RESPONSE
Ok, thank you. We have corrected the text.

Page 2, line 11-12: Change "by trees, through leaves stomata." to "by trees through leaf stomata."
RESPONSE
Ok, thank you. We have corrected the text.

Page 2, line 12: change "physiological factors which drive" to "physiological factors that drive".
RESPONSE

Ok, thank you. We have corrected the text.

Also, one line below, "availability which is positively..." to "availability that is positively…".
RESPONSE
Ok, thank you. We have corrected the text.

Page 2, line 16: change "many processes which have still to be understood in deep" to "many processes remaining to be understood in depth".
RESPONSE
Ok, thank you. We have corrected the text.

Page 2, line 16: please add Cape, 2009 to the reference list.
RESPONSE
Ok, thank you. We have added Cape 2009 to the reference list.

Page 2, line 22: change "Actually, very few studies" to "Actually, only a few studies" to fit better with the meaning of the sentence.
Page 2, line 23: change " forest, for example, " to " forest. For example, " to break a run on sentence to separate sentences.
Also, please correct the author's name: Launianen.
RESPONSE
Ok, thank you, corrected in Launiainen.

Page 2, line 24: I am not sure the statement "Dorsey et al., (2004) focused on the role of soil NO emission in the ozone flux dynamics." is accurate. I think the role of soil NO emission is only part of the discussion in that paper.
RESPONSE
Ok, the reviewer is right, the paper by Dorsey et al. 2004 was not focused only on the role of soil NO emission in the $O_3$ fluxes dynamics. The paper considered also the role of $NO_2$ fluxes at different levels in the forest profile. We have changed the sentence to clarify this.

Page 2, line 25: "There are still very few studies…" should be either "There are still few studies….", which basically means "There are no studies…", or "There are only a few studies….", which means there are some, but not many/not enough number of studies.
RESPONSE
Ok, thank you. We have corrected the text.

Page 2, line 27: Change "On the contrary, measurements of vertical gradients of ozone concentration have already been quite investigated (Fontan et al. 1992; Keronen et al. 2003; Utiyama et al. 2003; Gerosa et al. 2005)." to "On the contrary, vertical gradients of ozone concentration have already been extensively investigated (Fontan et al. 1992; Keronen et al. 2003; Utiyama et al. 2003; Gerosa et al. 2005)."
RESPONSE

Ok, thank you. We have corrected the text.

Page 7, line 29: Change "which are negligible respect to those related to the other considered." To "which are negligible with respect to those related to the other considered."
RESPONSE
Ok, thank you. We have corrected the text.

Also, it is not clear what "the other" is referring to.
RESPONSE
In this case "the other" is referred to the other scalar.

Page 8, line 5 and line 15, "per seconds" should be "per second".
RESPONSE
Ok, thank you. We have corrected the text.

Page 8, equation (2): I don't think SNO is correct here. With SNO, the unit of FNO would not be a unit for flux.
RESPONSE.
Thank you for pointing this out. The reviewer is right, equation (2) in the text was indeed wrongly reported and we apologize for that. The text and equations have been corrected.

Page 9, line 12, 14, 20: Please fix the figure numbers.
RESPONSE
Ok, thank you. The figure numbers have been fixed.

Page 9, line 28: It would be good to specify or characterize what "quite irregular" mean here. In other words: what do you mean by "quite irregular"?
RESPONSE
In this sentence with the term "quite irregular" we meant that the friction velocities ($u*$) above the canopy during the night did not show a small variation in values such as for example the $u*$ at 5 and 16 m. However, we understand that this term can be misleading. We have changed the sentence in order to clarify what we meant.

Page 11, line 23: "ozone fluxes and LE fluxes seem to be correlated". Would it be more illustrative to plot ozone fluxes vs. LE fluxes?
RESPONSE
Ok, thank you for this suggestion.
In the first submitted version of the paper these two Figures (ozone fluxes and LE fluxes) were reported together. Following this suggestion and the suggestions made by the reviewer #1 on the redistribution of figures between the main text and the supplementary material, we have rejoined the two Figures and moved them in the supplementary material as Figure S4(a) and S4(b).

Page 13, line 1: "only few studies" should be "only a few studies".
RESPONSE
Ok, thank you, we have corrected the text.

Page 13, line 6: Please add Arya, 1989 to the reference list.
RESPONSE
Ok, the year of this reference was wrong. The correct reference is Arya, 2001. We have corrected the text. The reference was already present in the reference list.

Page 31, Figure 9: Shouldn't the units for NOx fluxes be ug N/m^2/h? If it's per second, the reported fluxes would be too large.
RESPONSE
Thank you for this remark. The reviewer is right and we apologize for this mistake. We have changed the unit of Figure 9 (now Figure 7).

**LIST OF RELEVANT CHANGES**

The main changes interested:

- Few sentences of the Abstract

- Some sentences of the Introduction, which has been also expanded

- A formula was corrected in the section 2.8 of Material and Methods section few requested specifications has been added to the Results

- The discussion and conclusions sections have been improved in some parts according to the suggestions of the reviewers

- Error bars have been inserted in the Figures as requested

- Some figures have been moved to the supplementary material and vice versa, as requested by the reviewers

- Table S1 was moved from the Supplementary Material to the main text as requested by one of the reviewers.

- Some new references have been added to text and to the reference list.

All the changes have been clearly indicated in the responses given to each reviewer. Moreover, all the relevant changes have been highlighted the attached marked up version of the manuscript.

[revised manuscript text omitted]

---

## Author Response (AR3)

Reviewer #1

This paper is much improved in the third round of review, but it needs to be edited for English. Below are my specific comments that need to be addressed before publication.
RESPONSE
Thank you for the comments and the suggestions. This revised version of the manuscript was edited for English language by a professional service company (American Manuscript Editors). We are confident that now all the English problems have been fixed.
Please note that according to the suggestions received during the language editing process the title of the manuscript has been slightly changed ("Characterisation" was changed to "Characterization" and "5 levels" was changed to "five levels").

Page 2, Line 26: are the BVOCs oxidizing?
RESPONSE
We apologize for this mistake. Obviously, the BVOCs are not oxidizing compounds. On the contrary they are oxidized by $O_3$. We have removed the term from the text.

Page 3, Line 1: please cut extensively
RESPONSE
OK, done.

Page 12, Lines 1-2: I think "indicated" is too strong here. Please change to "suggested"
RESPONSE
OK, done.

Page 12, Lines 4-6: the smallest ozone or LE fluxes? How do the authors know the increase in ozone flux was due to increased stomatal activity? Seems to me like these findings are not too certain and should be expressed that way
RESPONSE
The sentence was referred to the smallest O3 fluxes, we have specified this in the text.
We see the reviewer's point: the increase of O3 fluxes could be due to an increase in the non-stomatal deposition together with an increase of stomatal conductance. According to the reviewer's suggestion we have rephrased the sentence in a more hypothetical form.

Page 12, Line 30: Please avoid the term "significant" if there is no statistical significance testing done
RESPONSE
Ok, thank you. We have rephrased the sentence.

Page 13, Line 28: Please avoid the term "obviously"
RESPONSE
Ok, done

Page 14, Lines 1-9: What is the point of the discussion of reaction between ozone and isoprene with respect to ozone fluxes? Isn't this reaction too slow to impact the ozone flux measurement? Are highly-reactive BVOCs thought to influence the ozone flux at this site? Nemitz et al. 2013 is not listed in the reference list and I cannot find it on Eiko Nemitz's Google Scholar.

RESPONSE

This part of the discussion was intended to discuss the possible role of the BVOCs in the enhancement of the O3 fluxes at 24 m.

The partition exercise made by Nemitz et al. (2013), which was based on the isoprene, by far the most emitted BVOC at the site, seems to suggest that the role of BVOCs in $O_3$ fluxes enhancement at 24 m was not so relevant.

However, the conclusions of Nemitz et al. could not be fully diriment on this issue, since the reaction between O3 and isoprene could be too slow to have an impact on the measured O3 fluxes, as highlighted by the reviewer.

Hence, we cannot exclude that fast reactions between O3 and undetected highly-reactive BVOCs occurred, e.g. with very reactive isoprenoids emitted at the canopy level such as B-Caryophillene, a sesquiterpene which reacts in the gas phase in few seconds. These reactions may have occurred for example in the sampling lines way before reaching the instrument inlet.

We have added a sentence in the discussion on the possibility that fast reactions between $O_3$ and undetected highly-reactive sesquiterpenes occurred. However, we have decided to cautiously keep this as a hypothesis because we cannot provide significant evidence of these compounds at Bosco Fontana.

Regarding the reference of Nemitz et al. (2013), this is related to a poster presented at the ACCENT 2013 meeting in Urbino (Italy) and summarized in the Book of Abstracts.

- Nemitz, E., Langford, B., Di Marco, C. F., Coyle, M., Braban, C., Twigg, M., Gerosa, G., Finco, A., Valach, A., Acton, J., Loubet, B., Schallart, S., Gasche, R., Diaz-Pines, E., Fares, S., Westerlund, J., Hallquist, Å., Gritsch, C., Zechmeister-Boltenstern, S., Sutton, M. A., 2013. Quantifying chemical interactions in a forest canopy – First results from the ÉCLAIRE campaign at Bosco Fontana, Po Valley. ACCENT-Plus Symposium, Urbino (Italy), 17-20 September 2013.

We did not include it in the reference list for an overlook. We apologize for that.

The reference list has been now updated.

We just realized that the book of abstract of the meeting is no longer available online, therefore we provide here a link to download the cited abstract:

https://www.dropbox.com/s/y0vj60vekt7vcup/Nemitz_session2_poster_Abstract.pdf?dl=0.

The poster can be required to the authors.

Page 15, Line 32: Soil should not be included here, right? Because the authors have isolated the "canopy" part of deposition.

RESPONSE

The reviewer is right, we have removed the term.

Page 15 Lines 29-33: Will the authors state in the text which hours of the day these numbers correspond to?

RESPONSE

Ok, we have inserted in the text a reference to the hours of the day.

Page 16, Lines 24-25: Same for abstract, I find this line confusing, and I think it's due to the grammar. Please separate the ozone reaction part of the sentence from the NO emission and deposition part of the sentence.

RESPONSE

OK, we have rephrased this sentence trying to better explain what we meant. The chemical $O_3$ sink at 24 m was responsible for the enhancement of $O_3$ fluxes measured at this level and was attributed to chemical reaction between $O_3$ and the NO that was emitted from the soil and deposited from the above atmosphere.

Page 17, Line 1-5: It's unclear what the authors are getting at here. Their first sentence makes me think that the authors are discouraging modelers from using this data. Is this the case?

RESPONSE

No, of course we are not discouraging modelers from using this data. On the contrary, we are confident that this data could be very useful for modelers, but at the same time we are aware that modelling the complex in-canopy gas trace dynamics on a daily scale could be very difficult, and in this sense it could represent a real challenge. Anyway, since we don't want to be misunderstood on this point we have removed the first sentence of this paragraph.

The figures are in much better shape. Sometimes there are some grey lines between subplots, or bordering certain subplots, and these should be removed.

RESPONSE

Ok, this was probably due to the conversion from the .doc format to the .pdf format. Anyway, we will provide the high-resolution version of each figure with no borders.

Figure 2: What are the units for the vertical distribution of vegetation?

RESPONSE

The green area represents the vertical distribution of the leaf area density and it was drawn to help the reader to identify the height of the upper and lower canopy layers, and the height of the understorey. The unit is $m^2$ of projected leaf surface/$m^3$ of air [$=m^2_{leaf}/(m^2_{soil}*m_{height}) = LAI/m_{height}$ ]. This unit has been reported in the caption of the Figure.

Figure 4: 100'000 should be 100,000; abs(L) should be $|L|$; Obukhov is spelled incorrectly

RESPONSE

Ok, thanks for the remarks, we have corrected the caption.

Figures 6-7: Why state here that we are only looking at the flux profile period? I thought this was the case for all figures.

RESPONSE

Ok, thank you, the reviewer is right and we have removed the term from the captions.

Figure 8: Specify what "MOD" means in caption text (I assume modified?)

RESPONSE
Ok, the meaning of MOD has been inserted in the caption.

[revised manuscript text omitted]

---

## Author Response (AR4)

Dear Dr. Gerosa,

I am pleased to accept your revised manuscript for publication in ACP. Thank you for your responses to the reviewers that improved this manuscript.
One issue that still needs to be addressed is the reference to Nemitz et al. 2013 which is not a document that is available to readers. Please replace this reference with one that is a suitable peer reviewed publication. After that you should upload your manuscript to complete the publication process in ACP.

RESPONSE

Dear co-editor,

the reference we have referred to is only available in a Book of Abstract of the meeting where the results were presented at (ACCENT 2013, Urbino). However a new publication related to this analysis is currently under development.
Hence , following your suggestion we have decided to remove the reference and to replace it as *Nemitz,2013 personal communication*.  In any case, since this part of the discussion is also supported by two others ACP publication (Schallahart et al.,2016; Acton et al, 2016), we believe we do not need to add other additional references.

With our best regards

Giacomo Gerosa and co-authors

[revised manuscript text omitted]